# Research

behaviour, ecology

rhinoceros, anti-poaching, drones, unmanned aerial vehicle, deterrents, behaviour

**Author for correspondence:**
Samuel G. Penny
e-mail: s.penny2@brighton.ac.uk

# Using drones and sirens to elicit avoidance behaviour in white rhinoceros as an anti-poaching tactic

Samuel G. Penny, Rachel L. White, Dawn M. Scott, Lynne MacTavish and Angelo P. Pernetta

School of Pharmacy and Biomolecular Sciences, University of Brighton, Brighton BN2 4GJ, UK

(iD) SGP, 0000-0002-7485-7897; APP, 0000-0003-4492-2798

Poaching fuelled by international trade in horn caused the deaths of over 1000 African rhinoceros (*Ceratotherium simum* and *Diceros bicornis*) per year between 2013 and 2017. Deterrents, which act to establish avoidance behaviours in animals, have the potential to aid anti-poaching efforts by moving at-risk rhinos away from areas of danger (e.g. near perimeter fences). To evaluate the efficacy of deterrents, we exposed a population of southern white rhinos (*C. simum simum*) to acoustic- (honeybee, siren, turtle dove), olfactory- (chilli, sunflower), and drone-based stimuli on a game reserve in South Africa. We exposed rhinos to each stimulus up to four times. Stimuli were considered effective deterrents if they repeatedly elicited avoidance behaviour (locomotion away from the deterrent). Rhinos travelled significantly further in response to the siren than to the honeybee or turtle dove stimulus, and to low-altitude drone flights than to higher altitude flights. We found the drone to be superior at manipulating rhino movement than the siren owing to its longer transmission range and capability of pursuit. By contrast, the scent stimuli were ineffective at inciting avoidance behaviour. Our findings indicate that deterrents are a prospective low-cost and *in situ* method to manage rhino movement in game reserves.

## 1. Introduction

The recovery of southern white rhino (*Ceratotherium simum simum*) populations to more than 20 000 individuals [1] from a remnant population of fewer than 50 breeding individuals at the end of the nineteenth century [2] is lauded as one of conservation's greatest successes [3]. However, this success is threatened by a rapid increase in rhino poaching [1] fuelled by a surge in demand from an increasingly affluent Southeast Asian market [4], where horn is used medicinally and as a symbol of status [5]. The rising costs of effective anti-poaching security are putting significant financial pressure on both national parks and private reserves [3], where the apprehension of poachers and reducing incursions are primarily achieved through foot and vehicle patrols [6]. There is thus a clear need to identify effective, low-cost, and readily applicable techniques to aid on-the-ground conservation efforts.

Poaching risk for rhino populations throughout their distribution is not homogeneous [1], being influenced by biophysical [7], geopolitical [8], and socio-economic factors [9]. Limited conservation resources are therefore focused on those areas exposed to the greatest levels of poaching risk [6]. Park *et al.* [10] found an area's poaching risk to be a function of its distance from the nearest water, buildings, vegetation, and roads, of the number of rhinos present, and of its topography. Spatio-temporal analyses of poaching patterns in African bush elephants (*Loxodonta africana*) show similar results, with the density of conspecifics, roads and rivers, condition of the vegetation, and distance from anti-poaching bases and boundaries all indicators of poaching risk [11–13].

Rhino poaching risk is also dependent on the time of day, and phase and position of the moon [14]. Twilight and night are the preferred time of poaching, particularly when they coincide with increased levels of lunar illumination [14] which will aid hunting but also poacher interdiction by rangers [6]. Poachers will also take advantage of bad weather conditions, which may limit the scope of patrols and increase their ease of escape [14].

The movement of rhinos away from these poaching hot-spots could be a useful anti-poaching tactic. Such a strategy would be most suited for use in private reserves, which are usually fenced and smaller than state or national parks [15,16] but hold approximately 30% of Africa's white rhino population and these animals are typically subject to more intensive management than national park populations [1]. Utilization of deterrents is one potential approach, which establishes avoidance behaviours in animals by exploiting defensive or anti-predator behavioural responses [17]. These behaviours are evoked through aversive or threatening stimuli that elicit fear or anxiety in the target subject, increasing real or perceived risk to a point where the costs of using a resource or area exceed its benefits [18]. To date, only one study has reported the use of deterrents in rhino management, in which electric fences were found to be effective at reducing crop raiding in Indian rhinos *Rhinoceros unicornis* [19].

In other species, successful deterrents employ non-physical structures such as sounds and smells to create 'metaphorical fences' [20]. White rhinos' disposition towards acoustic and olfactory disturbances [21] may mean they are also susceptible to these forms of deterrent. Acoustic deterrents can elicit a generalized threat response through loud or novel noises (e.g. bangs in rabbits [22]) or repel animals through pain and discomfort (e.g. artificial tones in seals [17]). The broadcast of such stimuli could exploit white rhinos' acute sense of hearing [23] to inhibit animal encroachments into specific areas. Other acoustic deterrents rely on conditioned responses towards unpleasant experiences [24]. White rhinos regularly disturb vegetation when rubbing against branches and moving through scrub [21], behaviour which may provoke defensive swarms of African honeybees (*Apis mellifera scutellate*) [25]. The broadcast of bee noise has the potential to incite a flight response in rhinos, if as occurs in African bush elephants *L. Africana* [24], individuals have experienced past aversive conditioning to stings. Olfactory stimuli such as capsaicin, an irritant present in chillies [26], have well-documented repelling effects across several species (e.g. in elephants [27,28], monkeys [29], and bears [30]). White rhinos have a highly developed sense of smell [31] and so exposure to noxious or novel scents could trigger avoidance behaviours through pain or neophobia. White rhinos will also flee from helicopters [32] and their preference for relatively open savannah grasslands [33] make habitats conducive to aerial pursuit. Hahn *et al.* [34] demonstrated how the disturbance effects of drones can be used to repel African bush elephants. Given that drones incite similar avoidance behaviours in a wide range of taxa (e.g. bears [35], seals [36], and birds [37]), the potential exists for them to incite a flight response in white rhinos.

To the best of our knowledge, the use of deterrents, that move animals away from areas of danger, remains unstudied in anti-poaching contexts. We therefore aimed to design and evaluate novel deterrent-based techniques that could be used in anti-poaching management approaches for white rhino conservation. Here, we investigate how white rhinos respond to acoustic, olfactory, and drone stimuli to determine their effectiveness as deterrents. Successful deterrents could be used to move rhinos from areas of high poaching risk to areas of refuge, providing a useful conservation tool for wildlife managers.

We predicted that exposure to certain stimuli would induce fear or anxiety in rhinos, inciting avoidance of the stimulus via a flight response. Mother–calf pairings were predicted to be more responsive to deterrents than either sub-adults or territorial bulls. For the three acoustic treatments, we tested the prediction that the noise of disturbed African honeybees would elicit a flight response, the noise of an oscillating siren would elicit an alert response, and the noise of territorial calls of Cape turtle dove (*Streptopelia capicola*) (from here on shortened to 'dove') would elicit no response. Rhinos were exposed to the approach of a drone flying at three different altitudes (less than or equal to 20, 60, and 100 m). We tested the prediction that rhinos would flee further from the lower altitude trajectories (less than or equal to 20 and 60 m), than from the high-altitude trajectory (100 m), where noise could be expected to be minimal and non-intrusive. For the olfactory stimuli, we tested the prediction that rhinos would demonstrate greater avoidance behaviour and reduced investigative behaviour to the scent of chilli oil than to the scent of sunflower oil.

## 2. Methods

Rhino behavioural responses were recorded following exposure to acoustic, olfactory, and drone-based stimuli between October 2016 and November 2017 on a population of dehorned white rhinos on a 47 km$^2$ private reserve in North West Province, South Africa. All experiments took place within bushveld savannah where grasses made up between 50 and 100% of the groundcover. Habitat type was standardized to avoid it influencing an animal's perception and response to risk [38]. If disturbance (vigilance towards the experimenter) occurred before the experiment began, then the experiment was delayed until rhinos settled back to their prior undisturbed behaviour. Prior to the start of acoustic and drone experiments, rhinos were identified via their unique ear notch patterns to prevent pseudo-replication. Rhinos were classed as subadults from maternal independence until they reached socio-sexual maturity. This is when males become solitary and/or territorial at 10–12 years old and at around 7 years old in females after the birth of their first calf [21]. Repeat experiments were conducted on the same individuals if a period of at least 24 h had elapsed since prior exposure. In mother–calf pairings, only mother behaviour was recorded. Rhinos were video recorded during exposure to the stimuli and any change in behavioural response was noted (table 1).

### (a) Acoustic deterrents

For the acoustic deterrent experiments, 12 rhinos were exposed to broadcasts of the bee, dove, and siren treatments up to four times each. The siren had a broad bandwidth to ensure a relatively high loudness, a spectral frequency within the range that rhinos vocalize [23], and a fast frequency modulation to maximize roughness [17]. The calls of a dove were selected as a control for the other two treatments owing to their ubiquitous occurrence and apparent neutral presence in the local soundscape. The bee and dove recordings were made on-site. Audio sequences were edited in Audacity (v. 2.1.1) and clipped to 60 s in length. To attenuate extraneous abiotic noise, the bee recording was low-pass filtered at 4500 Hz with a 6 dB per octave roll-off. The siren consisted of a repeated ascending tone; this consisted of a sine waveform rising in spectral

**Table 1.** Behavioural classifications and definitions used to measure rhino responsiveness towards a deterrent. Letters denote the trials for which behaviours are of relevance: acoustic (A); drone (D), and olfactory (O) deterrent. All behaviours marked by an asterisk were summed as a measure of awareness.

| behaviour | deterrent | definition |
|---|---|---|
| investigative* | A, D, O | locomotion (directed walking or running) towards the deterrent |
| | O | The sniffing or chewing of the deterrent |
| alert* | A, D, O | vigilance towards the deterrent (standing with the head held above the ground) |
| flight* | A, D, O | locomotion away from the deterrent (directed walking or running). Head held high, tail often curled |
| crossing | O | incidents of stepping over and past the rope |
| ignore | A, D, O | all other behaviours were classified as unresponsive, e.g. foraging. Alert behaviours were coded as unresponsive if they occurred before exposure, or if vigilance was towards another stimulus. Locomotive behaviour was not considered flight if it was undisturbed or not directed from the stimulus, e.g. walking during foraging |

frequency from 500 to 5000 Hz looped to a 2 Hz cycle. Sounds were broadcast louder than the recorded volume to compensate for speaker distance. The amplitude of the three sequences measured 86–66 dBC at 50–150 m distances in field conditions. This was similar to African honeybee playback experiments on elephants (66.1 dB at 10 m) [24]. Sounds were broadcast through two 30 W horn speakers (frequency range: 250 Hz–10 kHz; TOA Corporation) placed on the roof of a vehicle (2 m), facing towards the rhinos. Playbacks were started when rhinos were downwind, and between 50 and 150 m of the speakers.

Rhino behavioural responses were measured for the 1 min duration of the playback experiments (table 1). Observations were truncated at 1 min to ensure that rhinos remained visible throughout the experiment and to aid their comparability with data taken from the drone. The duration of investigative, alert, and flight behaviours was a measure of 'awareness' of the stimulus. The 'distance travelled' in response to the stimulus was a measure of flight response. The length of shorter distances (less than 10 s of movement) were estimated from rhino body length (approx. 3 m) relative to features in the video, for longer distances changes in rhino location were calculated via a range finder (Leica Rangemaster CRF 1600-R).

## (b) Drone deterrent

For the drone deterrent experiments, 12 rhinos were exposed to flights at low (less than or equal to 20 m), mid (60 m), and high (100 m) altitudes three times each. To avoid bias from condition order, the initial drone altitude was randomized with each subsequent exposure a different altitude to the preceding one. All drone experiments were performed with a multi-rotor DJI Mavic Pro. The drone was selected for its manoeuvrability, portability, availability as an off-the-shelf model, and its similarity to the drone models used to scare elephants [34].

The drone flights were initiated at least 150 m from the rhinos to avoid prior/post-experimental exposure. Following launch, the drone ascended to one of the three selected altitudes and flew in a straight, steady, level trajectory towards the epicentre of each rhino or rhino grouping. If the drone reached this overhead point, it hovered above the rhino for up to 5 min. If the rhinos moved, the drone pursued them for up to 1 min. The speed of the drone was kept to approximately $10\,\mathrm{m\,s^{-1}}$ throughout the experiment. The amplitude of the drone was measured from 1.5 m above the ground at three altitudes (76 dBc at 20 m; 67 dBc at 60 m, 61 dBc at 100 m) along with the peak spectral frequency, which at 6494 Hz is within rhino hearing range [23].

Rhino 'awareness' was recorded for a 1 min period following the first observed investigative, alert, or flight behaviour towards the drone (table 1). Rhino 'reaction distance' was recorded as the distance between the rhino and drone on the first observation of

awareness, if no response occurred, the closest distance reached between the rhino and drone (the drone altitude when hovering overhead) was recorded. To calculate this, a rhino's spatial location, recorded before launch, was subtracted from the drone's location, recorded every 10th of a second by an onboard GPS. Rhino flight response was quantified as the 'distance travelled' during a 1 min period following their first locomotive response to the drone (table 1). This was calculated by subtracting the difference between the rhinos start-, mid- (taken if the rhino stopped or changed direction), and end-positional coordinates following exposure to the drone. Mid- and end-positional coordinates were calculated from the position of the rhino in relation to the drone's location using the drone video output, internal compass, video timings, and satellite imagery (Sentinel 2, European Space Agency).

## (c) Olfactory deterrents

For the olfactory deterrent experiments, rhinos were exposed to ropes infused with chilli and sunflower oil. Accurate individual identification was not always possible and so responses were taken from a pool of 17 individuals with each exposure event treated as an independent data point. Thus, no tests of habituation were conducted. Chilli powder (specifically *Skopdonner*, a local cultivar of South African bird's-eye chilli, which scores around 50 000–175 000 Scoville heat units) was mixed with sunflower oil (1 : 10 ratio). A pure sunflower oil treatment was selected as a control. Lengths of 5 m natural fibre sisal rope were infused with scent by soaking them in one of the two treatment types for 24 h. Deployed ropes had scents reapplied after 5 days.

The lengths of scent-infused rope were laid across welltrodden animal trails that led to water bodies and showed recent signs of rhino activity. Rhino exposures were monitored by camera traps (Bushnell Trophy Cam) placed approximately 10 m away from the rope and 1.5 m high. Responses were recorded for the period that the rhino stayed within 5 m proximity of the rope (table 1). Avoidance of the stimulus was also determined by noting from recordings whether or not rhinos stepped over the scent stimulus.

## (d) Data analysis

Statistical analyses were conducted in R (v. 3.4.3) [39] to evaluate the effectiveness of each deterrent. For the acoustic and drone deterrents, the first set of analyses tested for differences in behavioural response between treatment types following a rhino's initial exposure to each stimulus. The second set of analyses tested for differences in behavioural response between replicates of each treatment type, as an indicator of habituation. Friedman's tests were used to account for the non-parametric distribution of

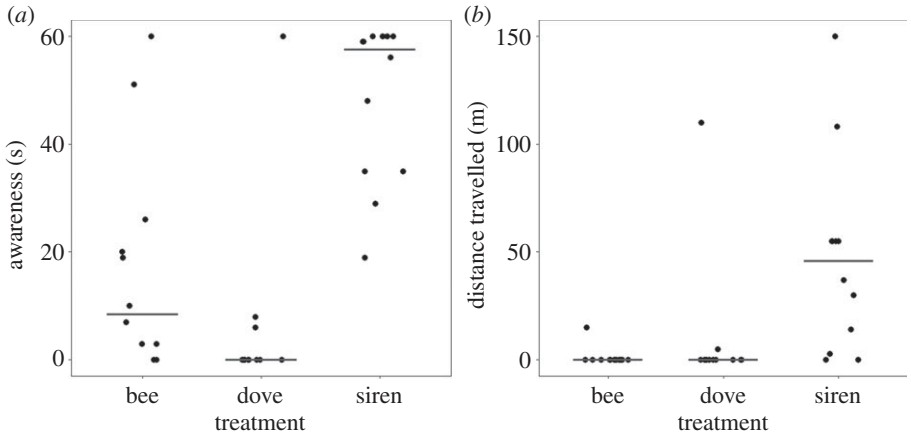

**Figure 1.** Rhino behavioural responses following initial exposure to each of the acoustic treatments for (a) awareness (duration of investigative, alert, and flight behaviours) ($n = 12$, obs. = 36) and (b) distance travelled ($n = 12$, obs. = 36). Data are horizontally jittered; lines show medians.

**Table 2.** Pairwise comparisons of rhino awareness duration and distance travelled in response to the acoustic deterrents. Analyses performed on responses with significant effects via Dunn's tests with Bonferroni corrections (per pair: $n = 12$, obs. = 24).

| parameter | $\chi^2$ | pairwise comparisons | | |
|---|---|---|---|---|
| | | siren × dove | siren × bee | dove × bee |
| awareness | 18.427 | <0.001 | 0.001 | 0.173 |
| distance travelled | 17.042 | <0.001 | <0.001 | 0.954 |

the data and the one-way repeated-measures designs, whereby each subject appeared in greater than one treatment and/or replicate. Dunn's tests with Bonferroni corrections were performed on significant results to establish any directions in trend and account for the family-wise error rate. For the olfactory deterrents, the absence of subject IDs precluded the use of a repeated-measures design. Consequently, a Mann–Whitney $U$ was used to test for differences in the duration of behaviours towards each treatment type and a $\chi^2$ test was used to establish the degree of independence between treatment type and behavioural counts. All analyses were two-tailed, and all $\alpha$ levels were set at 0.05.

## 3. Results

### (a) Acoustic deterrents

Significantly longer durations of awareness occurred in response to the siren (median = 57.5 s) than to either the bee (median = 8.5 s) or dove (median = 0 s) treatments (Friedman $\chi_2^2 = 15.591$, $p < 0.001$, $n = 12$, obs. = 36; figure 1 and table 2). The distances rhinos travelled also showed significant variation between acoustic treatments (Friedman $\chi_2^2 = 15.250$, $p < 0.001$, $n = 12$, obs. = 36; with rhinos moving significantly further in response to the siren (median = 46 m) than to either the bee (median = 0 m) or dove treatments (median = 0 m; table 2). When responding to the siren, subadults fled further (median = 55 m, $n = 4$) than both mother–calf pairs (median = 37 m, $n = 5$) and adult bulls (median = 3 m, $n = 3$).

Replicates of the siren resulted in no detectable change in awareness levels (Friedman $\chi_3^2 = 0.857$, $p = 0.835$, $n = 6$, obs. = 24) or distance travelled (Friedman $\chi_3^2 = 4.932$, $p = 0.177$, $n = 6$, obs. = 24) between experiments. Similarly, no changes were observed for either the dove (awareness—

Friedman $\chi_3^2 = 0.875$, $p = 0.832$, $n = 8$, obs. = 32; distance travelled—Friedman $\chi_3^2 = 3$, $p = 0.392$, $n = 8$, obs. = 32) or bee (awareness—Friedman $\chi_3^2 = 7.393$, $p = 0.060$, $n = 8$, obs. = 32; distance travelled—Friedman $\chi_3^2 = 5.857$, $p = 0.112$, $n = 8$, obs. = 32) treatments.

### (b) Drone deterrent

Rhinos could perceive the drone up to at least 100 m in altitude (electronic supplementary material, figure S1) and showed a near full minute of awareness to the initial drone experiments (figure 2). Rhino reaction distance and awareness to the initial drone experiments did not differ significantly between the three treatments (reaction distance—Friedman $\chi_2^2 = 3.455$, $p = 0.178$, $n = 11$, obs. = 33; awareness—Friedman $\chi_2^2 = 0$, $p = 1$, $n = 11$, obs. = 33). However, the distances rhinos travelled in response to the initial drone experiments differed significantly between the three treatments (distance travelled—Friedman $\chi_2^2 = 6.681$, $p = 0.035$, $n = 12$, obs. = 36). Rhinos moved significantly further in response to the drone flying at the low-altitude treatment (median = 61 m, $n = 12$; table 3) than they did to the high-altitude treatment (median = 10 m, $n = 12$), with the distance travelled in response to the mid-altitude treatment falling in between the two (median = 20 m, $n = 12$). Distance travelled was consistently high in mother–calf groupings (median low = 65 m, mid = 40 m, high = 45 m, $n = 5$), with greater levels of variation between treatments seen in subadult groupings (median low = 49 m, mid = 12.5 m, high = 0 m, $n = 4$) and adult males (median low = 67 m, mid = 20 m, high = 0 m, $n = 3$).

Several behaviour responses diminished following replicates of the drone stimuli (figure 3). Rhino reaction distance varied significantly following repeat exposure to the low-altitude treatment (Friedman $\chi_2^2 = 11.561$, $p = 0.003$, $n = 11$, obs. = 33)

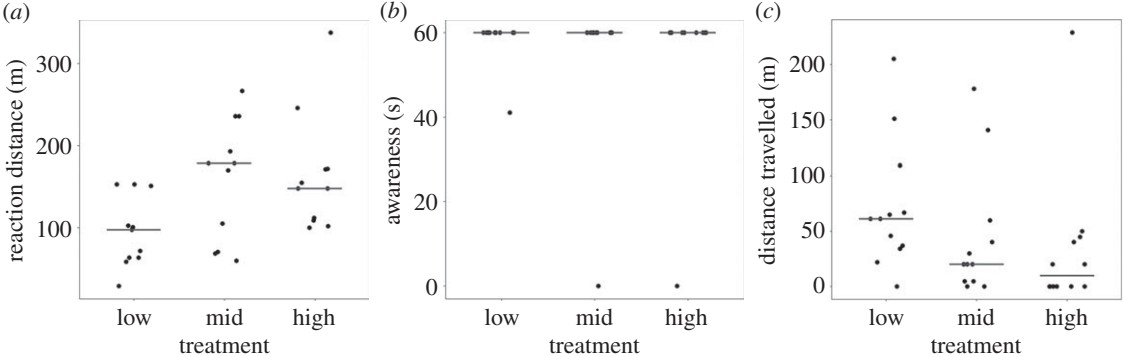

**Figure 2.** Rhino behavioural responses following initial exposure to each of the drone altitudes for (a) reaction distance (n = 11, obs. = 33), (b) awareness (duration of investigative, alert, and flight behaviours) (n = 11, obs. = 33), and (c) distance travelled (n = 12, obs. = 36). Data are horizontally jittered; lines show medians.

**Table 3.** Pairwise comparisons of rhino reaction distance, awareness duration, and distance travelled in response to drone flights at three altitudes. Analyses were performed on responses with significant effects after first exposure (between treatments) and repeat exposure (within treatments) via Dunn's tests with Bonferroni corrections. Sample sizes and observation numbers are listed per pair in subscript.

| parameter | | $\chi^2$ | pairwise comparisons | | |
|---|---|---|---|---|---|
| first exposure | | drone | low × mid | low × high | mid × high |
| drone | distance travelled$_{(12,24)}$ | 7.269 | 0.013 | 0.101 | 0.634 |
| repeat exposure | | drone | 1st × 2nd | 2nd × 3rd | 1st × 3rd |
| low | reaction distance$_{(11,22)}$ | 7.950 | 0.066 | 0.903 | 0.017 |
| mid | reaction distance$_{(10,20)}$ | 8.175 | 0.025 | 1.000 | 0.016 |
| mid | awareness$_{(10,20)}$ | 11.204 | 0.141 | 0.141 | 0.001 |
| high | awareness$_{(11,22)}$ | 9.310 | 0.075 | 0.442 | 0.004 |
| high | distance travelled$_{(12,24)}$ | 5.314 | 0.044 | 0.995 | 0.122 |

and mid-altitude treatment (Friedman $\chi^2_2 = 9.657$, $p = 0.008$, $n = 10$, obs. = 30), but not the high-altitude treatment (Friedman $\chi^2_2 = 4.667$, $p = 0.097$, $n = 11$, obs. = 33), with reaction distance declining over time (table 3 and figure 3). Awareness towards the stimuli did not vary significantly in response to replicates of the low-altitude treatment (Friedman $\chi^2_2 = 2.10$, $p = 0.350$, $n = 11$, obs. = 33). However, significant changes in awareness were detected after replicates to the mid-altitude (Friedman $\chi^2_2 = 11.438$, $p = 0.003$, $n = 10$, obs. = 30) and high-altitude treatments (Friedman $\chi^2_2 = 9.920$, $p = 0.007$, $n = 11$, obs. = 33); decreased levels of awareness were apparent for the later replicates (table 3). Despite these drops in awareness and reaction distance, rhinos travelled a similar distance across replicates of the low-altitude (Friedman $\chi^2_2 = 5.070$, $p = 0.079$, $n = 11$, obs. = 33) and mid-altitude treatment (Friedman $\chi^2_2 = 3.706$, $p = 0.157$, $n = 12$, obs. = 36). However, distances travelled in response to the high-altitude treatment did show significant variation between replicates (Friedman $\chi^2_2 = 11.20$, $p = 0.004$, $n = 12$, obs. = 36); with the greatest difference between the first and second replicates (table 3 and figure 3).

## (c) Olfactory deterrents

Awareness towards the olfactory deterrent did not differ significantly between the chilli and sunflower oil treatments ($W = 873$, $p = 0.255$, $n = 78$), with sniffing, chewing, and alert behaviours observed towards both treatment types (figure 4). No association was found between the tendency

of a rhino to cross over a rope following their approach of it and scent treatment ($\chi^2_1 = 0.915$, $p = 0.339$, $n = 78$). Thus, following approach of the rope, most rhinos continued to travel along the game trail, crossing over the olfactory deterrents (electronic supplementary material, figure S1).

## 4. Discussion

Exposure to both the siren and low-altitude drone treatment repeatedly elicited a flight response from the studied rhino, enabling the movement of animals away from undesirable areas. Although the distances travelled were short, in many cases, the rhinos continued to flee after observations had ended; on two occasions, rhinos ran over 500 m in response to the drone, even without pursuit, and on four occasions over 250 m from the siren. While previous studies with elephants have shown both the olfactory stimulus of chilli powder [37] and auditory stimulus of bee noise [24] to be effective deterrents, the results of this study show neither to be effective for rhinos.

Rhinos may soon return to an area if the costs of avoiding the stimulus are outweighed by the benefits of staying put and foraging [40]. Thus, the chances of return could be lowered if higher quality areas of habitat are maintained in other more suitable areas of a reserve. Deterrents may be less effective in periods of reduced resource availability, such as drought, when rhinos have a more limited choice in grazing

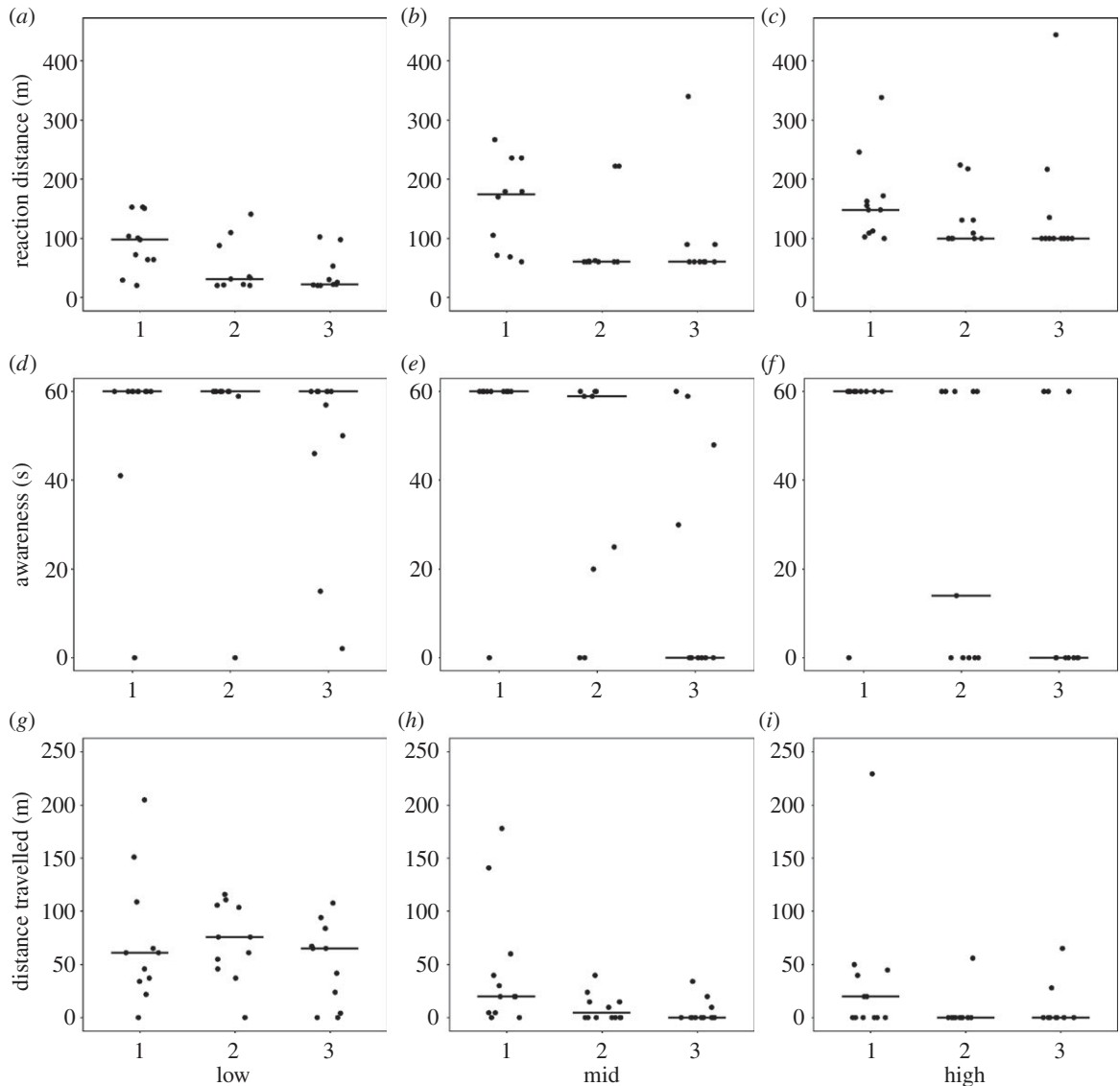

**Figure 3.** Rhino behavioural responses to the drone deterrent following three replicates per rhino at each altitude: (*a–c*) reaction distance, (*d–f*) awareness (duration of investigative, alert, and flight behaviours) and (*g–h*) distance travelled. Data are horizontally jittered; lines show medians (*a,c,d,f*: $n = 11$, obs. = 33; *b,e*: $n = 10$, obs. = 30; *g–i*: $n = 12$, obs. = 36).

areas [33]. Dominant males will also be more inclined to return to an area than other social classes, given their need to regularly patrol and demarcate their territories [21,31].

The sample sizes (less than or equal to 17 rhinos per analysis) are comparable to existing studies of deterrents [41–43] and reflect the difficulty of exposing free-ranging mammals to experimental stimuli and the need to minimize undue stress [44]. Furthermore, unlike several previous studies [27,34,42], the repeated-measures design of the acoustic and drone analyses provides a robust control for individual variation. Perhaps more importantly, the study population and field site are representative of those found in other private reserves [16] where the deterrents will have the greatest conservation impact.

## (a) Acoustic deterrents

The rhinos did not appear to perceive the risk from the bees as great enough to initiate flight behaviour [40]. This contrasts with the growing body of evidence for their use with African bush elephants [24–25,45], perhaps because the thick skin of rhinos, adapted to shield against attacks from conspecifics, provides sufficient protection against aggressive swarms of bees

[46]. Given the importance of sound in rhino communication and their perception of changes within their environment [21,23], the repeated avoidance of the siren suggests rhinos responded to its roughness, with the fast frequency modulation inducing a psychophysiological unpleasantness [17].

## (b) Drone deterrent

The levels of acoustic and visual disturbance caused by the drone are a function of its proximity to the rhino, with the drone becoming louder and more intrusive as it approaches. As rhinos reacted to the drone when facing away from the drone's angle of approach, acoustic output alone can be enough to initiate a response. Mother–calf pairings fled from all three altitudes of the drone, suggesting they may perceive risk differently to other social groupings and be more susceptible to the deterrent than solitary males or sub-adult groupings. Individual variation in traits such as sex or body size influence trade-offs between the avoidance of perceived risk and fitness-enhancing activities [38,41]. For example, males of both Asian elephants (*Elephas maximus*) and mountain beavers (*Aplodontia rufa*) are less susceptible following exposure to aversive stimuli than females [27,41].

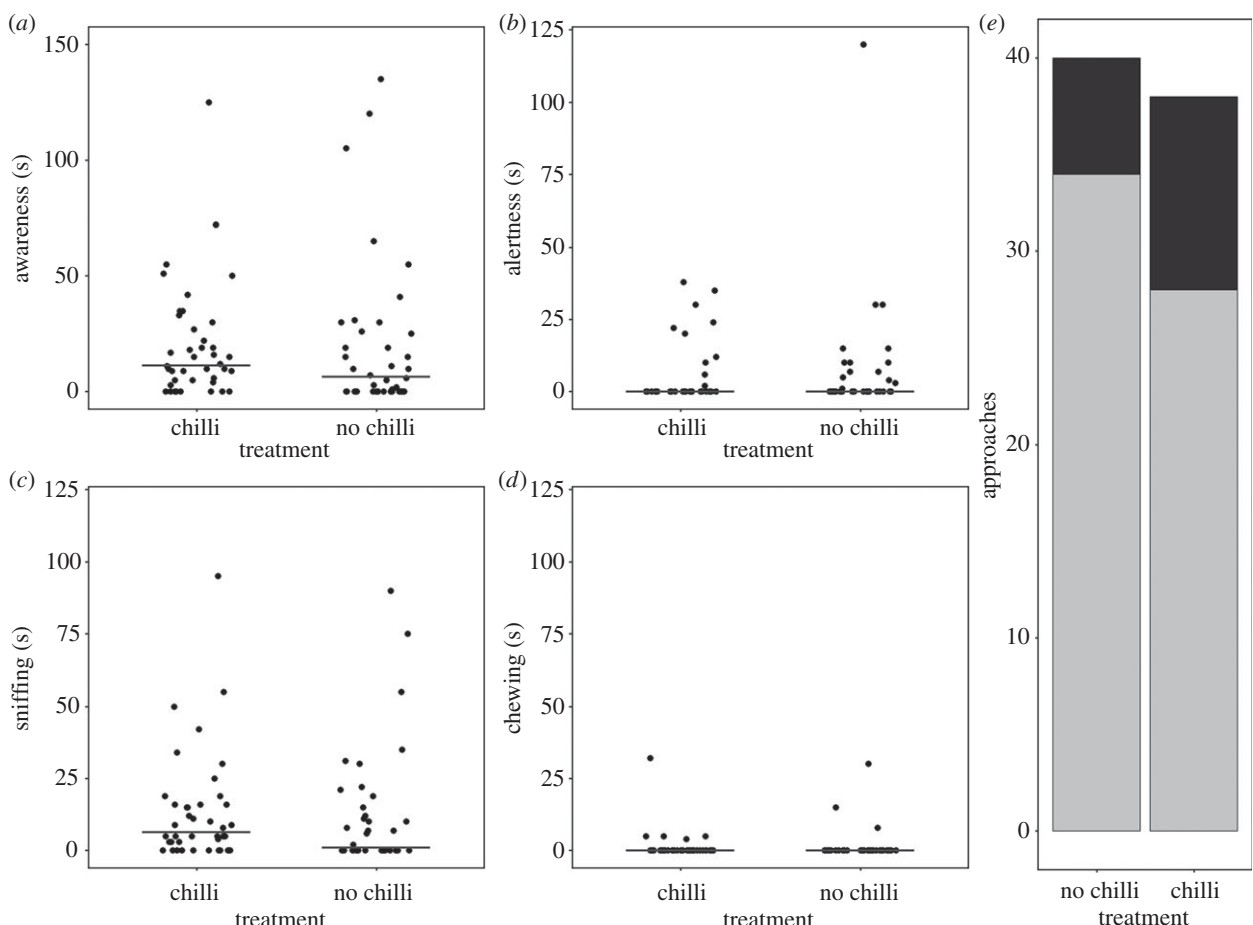

**Figure 4.** Rhino behavioural responses following exposure to the olfactory deterrents showing: (*a*) awareness, and behavioural subsets of awareness: (*b*) alertness, (*c*) sniffing, and (*d*) chewing, whereby data are horizontally jittered and lines show medians; (*e*) tendency to cross the rope, whereby light and dark shading indicate a crossing and no crossing, respectively (*n* = 78).

In these cases, the males appear to be less risk-averse owing to their comparatively greater body size and/or greater energy requirements [27,41], with females acting more cautiously owing to their accompaniment of juveniles [41,47]. Rhino mother–calf pairings may thus show differing perceptions of risk and a stronger response to deterrents than either adults or subadults owing to their calves' greater susceptibility to predation risk [21].

A greater level of exposure was necessary to induce a behavioural response at low- and mid-altitude treatments, as shown by decreases in reaction distance in the later replicates. Diminishing rates of awareness in the mid- and high-altitude replicates indicate rhinos perceived the drone to decline in threat. Although there was no change in awareness in response to the low-altitude treatment, the concurrent decrease in reaction distance meant that greater levels of acoustic or visual exposure to the drone were necessary to maintain a similar degree of responsiveness. Rhinos continued to travel the same distance across replicates of the low- and mid-altitude treatments but reduced to near zero at the high-altitude treatment, indicating rhinos habituate quickly to nominal drone exposure but continue to flee from more intense levels of exposure. It remains possible that the distance travelled from the drone would diminish across all altitudes following further replicates. However, as rhinos fled from the drone after a cumulative nine replicates, the deterrent can work without a significant reduction in effect at least at this level of exposure. As well as total exposure, the frequency of exposure can influence habituation rates

[48]. Diminished responses may recover fully if the stimulus is withheld over time, in what is known as spontaneous recovery [48]. A less frequent exposure rate than that used in the study (9 times over 90 days) may see responses maintained over a longer period. Minor changes to a signal may be enough to restore the original behavioural response [49], with exposure to a single strong or different stimulus leading to dishabituation [48]. To prolong the effectiveness of the drone, exposure can be intensified by flying at lower altitudes than that trialled in the study, limited to within a few metres of the rhino, or by flying at faster speeds to increase the level of noise output and reduce decision-making time [38].

## (c) Olfactory deterrents

Neither of the olfactory treatments were successful in controlling rhino movement. Investigative sniffing and chewing behaviours showed rhinos could perceive the stimuli, but neither substance was aversive or appeared to be causative of pain or irritation. As rhinos paused to investigate the treatments, the stimuli encouraged rhino to stay within their vicinity for longer, with both treatments acting as an attractant. Hedges & Gunaryadi [50] failed to elicit an aversive response to a chilli rope deterrent in Asian elephants and argued that the reported successes of similar deterrent studies may have been owing to their parallel usage of other deterrents such as increased levels of farmer vigilance [28,51]. In studies of human–wildlife conflict mitigation, robust factorial designs are not always possible, as the failure

of non-effective controls can have a direct impact on people's livelihoods [42]. It remains possible that rhinos could show a response to the aerosol deployment of chilli, but as the application of sprays has a greater potential to cause undesirable symptoms such as apnoea and temporary blindness [26,30], they are less suitable for exploratory use.

## (d) Conservation implications

In conclusion, in addition to identifying abiotic auditory stimuli as effective deterrents, our research is the first to identify the potential of drones as a management tool for active movement of rhinos in protected areas. By using their disturbance effects, we have found results contrary to those of Mulero-Pazmany et al. [52] who reported no rhino 'alarm reaction or flight responses' to reconnaissance flights at altitudes between 100 and 180 m. By reducing the altitude of flights, we have found a technique whereby reserve managers can use drones to readily respond to reports of at-risk animals. Pursuit by the drone is only limited by the model's transmission range and battery life, which are much greater than the short periods tested in the experiment. Drones require no ground-based infrastructure or nearby operators and can be flown into any position regardless of terrain and vegetation type. Drone deterrents would be most applicable to small private reserves, where rhinos have access to perimeter zones or exposed areas, particularly during heightened periods of risk (e.g. around the full moon [14] or when poaching syndicates are known to be operating in the area [52]). They are less suited for use in larger state or national parks with semi-porous borders and near-constant poaching activity [4]. Furthermore, owing to their additional surveillance functions, poachers are likely to be incentivized to avoid areas where they operate. For anti-poaching units that already use drones for surveillance purposes, there is no additional outlay in equipment costs, with their use as a deterrent adding an additional function to reconnaissance [52].

Data accessibility. Data available from the Dryad Digital Repository: https://doi.org/10.5061/dryad.tb21f30 [53].

Authors' contributions. S.G.P., A.P.P., R.L.W., and D.M.S. conceived and designed the study. S.G.P., L.M., and R.L.W. collected the data. S.G.P. analysed the data. S.G.P. and A.P.P. wrote the paper. R.L.W. and D.M.S. reviewed the manuscript. All authors gave final approval for publication.

Competing interests. We declare we have no competing interests.

Funding. This work was supported by the Earthwatch Institute as part of the Conserving Endangered Rhinos in South Africa programme. S.G.P. was additionally supported by a University of Brighton doctoral studentship.

Acknowledgements. Thank you to D. MacTavish without whom this project would not have been possible. We also express our sincere thanks to the reserve staff and volunteers who supported and assisted us in the field, particularly those from Earthwatch and the Nkombi team.

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
