## [Reviewer comments · Proceedings of the Royal Society B: Biological Sciences]

Review History

RSPB-2019-0240.R0 (Original submission)

Review form: Reviewer 1

Recommendation

Major revision is needed (please make suggestions in comments)

Scientific importance: Is the manuscript an original and important contribution to its field?

Excellent

General interest: Is the paper of sufficient general interest?

Excellent

Quality of the paper: Is the overall quality of the paper suitable?

Acceptable

Is the length of the paper justified?

Yes

Should the paper be seen by a specialist statistical reviewer?

No

Do you have any concerns about statistical analyses in this paper? If so, please specify them explicitly in your report.

No

It is a condition of publication that authors make their supporting data, code and materials available - either as supplementary material or hosted in an external repository. Please rate, if applicable, the supporting data on the following criteria.

Is it accessible?

Yes

Is it clear?

Yes

Is it adequate?

Yes

Do you have any ethical concerns with this paper?

No

Comments to the Author

The authors present important data that could be used to help protect rhinos from poaching risk. I think this study is ultimately worthy of publication; the data are sound and presented well, and the topic is pressing. My concern is that the study is framed almost entirely without ecological or behavioural context. I think the paper has the potential to meet the outstanding standards of Proc B if the authors undertake the major revision to restructure, and to a some extent, re-write their paper.

The authors make some strides at providing ecological/behavioural context in the discussion to help explain the results. However, this is belated and insufficient. The authors need to provide more robust justifications for the deterrents they chose, as it relates to the lived experiences of the rhinos. For example, the authors write (starting on line 108), "For the olfactory stimuli, we tested the prediction that rhinos would demonstrate greater avoidance behaviour and reduced investigative behaviour to the scent of chilli oil than to the scent of sunflower oil, due to the potential irritating properties of chilli." The justification here is devoid of its meaning as it relates to what we know of rhinos.

Furthermore, the authors (starting on line 82) make the very case for the need to provide greater ecological and behavioral information – for the species overall, as well as for the particulars of the study. The authors write, "Poaching risk for rhino populations throughout their distribution is not homogenous [5], being influenced by biophysical [21], geopolitical [22] and socioeconomic factors [23]. Therefore, even at a protected area scale, poaching rate will vary spatially, with poaching hotspots typically centred around geographical features such as waterholes [24] and boundary features [25-26], leading to uneven distribution of anti-poaching efforts [24]." Thus, if any potential deterrents, such as the drones, are to be effectively and situationally deployed to reduce poaching risk, practitioners/managers need to fully understand the results of this study in its broader context. For example, given the complexity of factors under what circumstances would drones be effective. What factors would render such a deterrent useless against poaching threat?

Overall, the paper devotes itself to presenting the data, while taking for granted the information (behavioural/ecological/geographical, etc.) that would actually give more heuristic credence to the findings. I look forward to seeing this valuable study presented in the framework it deserves.

Review form: Reviewer 2

Recommendation

Accept with minor revision (please list in comments)

Scientific importance: Is the manuscript an original and important contribution to its field?

Excellent

General interest: Is the paper of sufficient general interest?

Excellent

Quality of the paper: Is the overall quality of the paper suitable?

Excellent

Is the length of the paper justified?

Yes

Should the paper be seen by a specialist statistical reviewer?

No

Do you have any concerns about statistical analyses in this paper? If so, please specify them explicitly in your report.

No

It is a condition of publication that authors make their supporting data, code and materials available - either as supplementary material or hosted in an external repository. Please rate, if applicable, the supporting data on the following criteria.

Is it accessible?

No

Is it clear?

N/A

Is it adequate?

N/A

Do you have any ethical concerns with this paper?

No

Comments to the Author

This study brings new information on the potential use of acoustic and olfactory deterrents in anti-poaching efforts to move rhinos away from high-risk areas. The manuscript is well-written. The authors came up with a good idea and went in the footsteps of previous research done mainly on elephants, which showed that various olfactory and acoustic stimuli or unmanned

aerial vehicle can deter them from entering unwanted areas (e.g. when wanting to raid crops). The authors tested if these deterrents could also be used to deter rhinos from areas which are considered high-risk because of poaching.

The study was well-designed although it would also be nice to see if the authors had any information on how long approximately the effect of the siren and unmanned aerial vehicle on the rhinos lasted. The authors mentioned that the trials were not repeated on the same individuals sooner than in 24 hours. Does this mean that the individuals can return to the same area at least in a day? In addition, the distance to which the rhinos ran away from the siren was not large (median 46m for all rhinos altogether, but only 3m for adult bulls). This distance is short taking into account that the poachers can hit anytime and it is not such a problem for them to move and search for the rhinos. However, the drones have a potential to pursue the rhinos for longer distances and could be used to get them out of high-risk areas in smaller reserves during the most sensitive time like e.g. around the full moon which the poachers are known to prefer (e.g. Mulero-Pazmany et al. 2014).

Here are some specific comments and a few suggestions that may improve the manuscript:

I think that the name of the article should be changed as the authors did not test if the drones and sirens reduce the risk of poaching. They tested if the rhinos move away from the drones and the sound of siren, but it remains unknown whether this can reduce the risk of poaching especially since it seems that the rhinos can come back to the same area soon and the distance to which they move away from the siren is short. In this line, I also suggest to re-phrase the last sentence of the abstract (line 40-42).

Line 146-152: You write that the playbacks were played when the rhinos were in distances 50-150 m from the speaker. It seems that the loudness of the playbacks was not changed depending on various distances of rhinos from the speakers? Do you think that this could have had any effect on the reactions of rhinos? That those who were closer reacted more intensively?

Line 150: I am missing the information about the manufacturer and parameters of the horn speakers. Their frequency range, etc.

Line 153: Could you, please, include here why you did not continue recording the rhino behaviour after the end of the playback? It would be interesting to know the reactions of rhinos even after the playback ended and to see what effect it had on them. Did the rhinos always stop reacting with either awareness / moving away already during the playback? Or was it impossible for you to continue with the observations due to e.g. the subjects getting out of sight?

Line 158-159: range finder info? Manufacturer, etc. ?

Line 243-245: Since you also analysed the data based on sorting out the individuals into sex-age classes, there should be an explanation and a citation (e.g. Owen-Smith (1973)) in the methods how you identified individuals as adults/subadults. In bulls, this could differ depending on if you regard them as adults based either only on their sexual maturity or on their socio-sexual maturity when they become solitary between the age 10-12 years.

Figures: Please, describe in captions what the boxes and the measures of central tendency represent.

Figures 1, 3, 4, 5: Some of the medians cannot be seen due to the black boxes. Can you, please, make the boxes lighter so the medians could also be seen in the figures?

Decision letter (RSPB-2019-0240.R0)

25-Mar-2019

Dear Mr Penny,

Thank you for providing the additional information requested about the sample sizes in your submitted paper "Drones and sirens reduce poaching risk by eliciting avoidance behaviour in southern white rhinoceros (*Ceratotherium simum simum*)"; we appreciated the thoroughness of your response. As already set out in our earlier correspondence, we were all agreed that this is a very interesting paper on an important topic, but we were concerned about the lack of exact information on sample sizes, and the indication of very low numbers of study species. The additional information you provided was useful for clarifying that, including the comparison with other comparable studies.

Having discussed the paper further with the Associate Editor who is handling it and also the Editor-in-Chief, we have decided on a Reject-Resubmit decision that gives you the opportunity to address these issues. In addition to the referees' and Associate Editor's comments below, please see my specific comments with regards to the presentation of the data and the statistics. However we also appreciate the potential sensitivity of the population information given the study species, so please get back to us if the request for full information on sample sizes is going to be problematic. Obviously we do not want to in any way endanger any animals, though I note that the rhinos are de-horned, which I hope might reduce any risk, and that the geographic location information is minimal.

The paper is therefore being rejected in its current form, but we would be happy to consider a resubmission, provided the comments of the referees are fully addressed. However please note that this is not a provisional acceptance.

We look forward to seeing your revised version.

Yours sincerely,
Loeske Kruuk

Editor

Proceedings B

Editor Comments (Loeske Kruuk)

As per the Associate Editor's comments as well, please clarify the sample sizes of numbers of animals involved in the different trials. Please also explain how the statistical tests used account for the non-independence of repeated measures on the same animals.

The figures also need to give an accurate indication of the data-sets: bar-charts of the form shown are not a helpful way to represent very small samples, but you could add the actual data values on top. (Have a look at Weissgerber et al. 2015. Beyond Bar and Line Graphs: Time for a New Data Presentation Paradigm. PLOS Biology 13:e1002128).

Finally, there needs to be some acknowledgement and discussion of the limitations of the small sample size, which obviously can include a reminder as to why sample sizes are far smaller than typical behavioural studies.

Associate Editor

Comments to Author:

After receiving two peer reviews, I continue to be impressed by the potential importance of the results presented in the MS: "Drones and sirens reduce poaching risk by eliciting avoidance behaviour in southern white rhinoceros (*Ceratotherium simum simum*)". Understanding mechanisms to reduce poaching risk are of pressing importance and have the potential to deepen knowledge about avoidance and deterrence behavior more generally. However, along with both reviewers, I found the framing the study to be somewhat limited in scope and scalability.

In addition to the excellent broad points raised by both reviewers regarding contextualizing, framing, and consideration of the poaching impact of the research, I believe that the work would also benefit from the following considerations:

- 1) The sample size, study subjects, and statistical approach were not properly explained (e.g., how many rhinos were studied should be in the methods; what is the unit of analysis for the models?). Moreover, the statistical methods contain at least one error/typo: The last sentence of the methods reads "all alpha levels were set at 0.05 MULTIPLIED by the number of hypotheses tested." I believe the authors meant to write "...DIVIDED by...".
- 2) Several, more minor issues: (a) I think that the use of multiple acronyms interferes with rather than facilitates comprehension. I would like to see all treatment conditions spelled out in the text. (b) Figure legends should contain more information in general, especially about treatments as well as descriptions of the unit of analysis and the N.
- 3) A detailed response to all the reviewers additional, specific concerns is also needed (e.g., approval by an Ethics Board and if not, please justify).

If the authors are able to address these concerns, I believe it could be a valuable and unique contribution to the literature.

Reviewer(s)' Comments to Author:

Referee: 1

Comments to the Author(s)

The authors present important data that could be used to help protect rhinos from poaching risk. I think this study is ultimately worthy of publication; the data are sound and presented well, and the topic is pressing. My concern is that the study is framed almost entirely without ecological or behavioural context. I think the paper has the potential to meet the outstanding standards of Proc B if the authors undertake the major revision to restructure, and to a some extent, re-write their paper.

The authors make some strides at providing ecological/behavioural context in the discussion to help explain the results. However, this is belated and insufficient. The authors need to provide more robust justifications for the deterrents they chose, as it relates to the lived experiences of the rhinos. For example, the authors write (starting on line 108), "For the olfactory stimuli, we tested the prediction that rhinos would demonstrate greater avoidance behaviour and reduced investigative behaviour to the scent of chilli oil than to the scent of sunflower oil, due to the potential irritating properties of chilli." The justification here is devoid of its meaning as it relates to what we know of rhinos.

Furthermore, the authors (starting on line 82) make the very case for the need to provide greater ecological and behavioral information – for the species overall, as well as for the particulars of the study. The authors write, "Poaching risk for rhino populations throughout their distribution is not homogenous [5], being influenced by biophysical [21], geopolitical [22] and socioeconomic factors [23]. Therefore, even at a protected area scale, poaching rate will vary spatially, with poaching hotspots typically centred around geographical features such as waterholes [24] and boundary features [25-26], leading to uneven distribution of anti-poaching efforts [24]." Thus, if any potential deterrents, such as the drones, are to be effectively and situationally deployed to reduce poaching risk, practitioners/managers need to fully understand the results of this study in its broader context. For example, given the complexity of factors under what circumstances would drones be effective. What factors would render such a deterrent useless against poaching threat?

Overall, the paper devotes itself to presenting the data, while taking for granted the information (behavioural/ecological/geographical, etc.) that would actually give more heuristic credence to the findings. I look forward to seeing this valuable study presented in the framework it deserves.

Referee: 2

Comments to the Author(s)

This study brings new information on the potential use of acoustic and olfactory deterrents in anti-poaching efforts to move rhinos away from high-risk areas. The manuscript is well-written. The authors came up with a good idea and went in the footsteps of previous research done mainly on elephants, which showed that various olfactory and acoustic stimuli or unmanned aerial vehicle can deter them from entering unwanted areas (e.g. when wanting to raid crops). The authors tested if these deterrents could also be used to deter rhinos from areas which are considered high-risk because of poaching.

The study was well-designed although it would also be nice to see if the authors had any information on how long approximately the effect of the siren and unmanned aerial vehicle on the rhinos lasted. The authors mentioned that the trials were not repeated on the same individuals sooner than in 24 hours. Does this mean that the individuals can return to the same

area at least in a day? In addition, the distance to which the rhinos ran away from the siren was not large (median 46m for all rhinos altogether, but only 3m for adult bulls). This distance is short taking into account that the poachers can hit anytime and it is not such a problem for them to move and search for the rhinos. However, the drones have a potential to pursue the rhinos for longer distances and could be used to get them out of high-risk areas in smaller reserves during the most sensitive time like e.g. around the full moon which the poachers are known to prefer (e.g. Mulero-Pazmany et al. 2014).

Here are some specific comments and a few suggestions that may improve the manuscript:

I think that the name of the article should be changed as the authors did not test if the drones and sirens reduce the risk of poaching. They tested if the rhinos move away from the drones and the sound of siren, but it remains unknown whether this can reduce the risk of poaching especially since it seems that the rhinos can come back to the same area soon and the distance to which they move away from the siren is short. In this line, I also suggest to re-phrase the last sentence of the abstract (line 40-42).

Line 146–152: You write that the playbacks were played when the rhinos were in distances 50–150 m from the speaker. It seems that the loudness of the playbacks was not changed depending on various distances of rhinos from the speakers? Do you think that this could have had any effect on the reactions of rhinos? That those who were closer reacted more intensively?

Line 150: I am missing the information about the manufacturer and parameters of the horn speakers. Their frequency range, etc.

Line 153: Could you, please, include here why you did not continue recording the rhino behaviour after the end of the playback? It would be interesting to know the reactions of rhinos even after the playback ended and to see what effect it had on them. Did the rhinos always stop reacting with either awareness / moving away already during the playback? Or was it impossible for you to continue with the observations due to e.g. the subjects getting out of sight?

Line 158-159: range finder info? Manufacturer, etc. ?

Line 243-245: Since you also analysed the data based on sorting out the individuals into sex-age classes, there should be an explanation and a citation (e.g. Owen-Smith (1973)) in the methods how you identified individuals as adults/subadults. In bulls, this could differ depending on if you regard them as adults based either only on their sexual maturity or on their socio-sexual maturity when they become solitary between the age 10–12 years.

Figures: Please, describe in captions what the boxes and the measures of central tendency represent.

Figures 1, 3, 4, 5: Some of the medians cannot be seen due to the black boxes. Can you, please, make the boxes lighter so the medians could also be seen in the figures?

Author's Response to Decision Letter for (RSPB-2019-0240.R0)

See Appendix A.

RSPB-2019-1135.R0

Review form: Reviewer 1

Recommendation

Accept with minor revision (please list in comments)

Scientific importance: Is the manuscript an original and important contribution to its field?

Excellent

General interest: Is the paper of sufficient general interest?

Excellent

Quality of the paper: Is the overall quality of the paper suitable?

Good

Is the length of the paper justified?

Yes

Should the paper be seen by a specialist statistical reviewer?

No

Do you have any concerns about statistical analyses in this paper? If so, please specify them explicitly in your report.

No

It is a condition of publication that authors make their supporting data, code and materials available - either as supplementary material or hosted in an external repository. Please rate, if applicable, the supporting data on the following criteria.

Is it accessible?

Yes

Is it clear?

Yes

Is it adequate?

Yes

Do you have any ethical concerns with this paper?

No

Comments to the Author

I was pleased to read the revised, much improved MS. I look forward to seeing it published after some further revision. My principal concern still holds: I'm still not seeing a clear, direct connection between rhino biology/physiology, behaviour/ecology and the use of the deterrents (e.g., lines 119-156). It still reads a bit vague. This remains true of the discussion as well, where I see more of a reiteration of results rather than a discussion as to rhino-specific reasons why certain results were observed (e.g., see discussion section on acoustic deterrents). That being said, the authors do include information to explain some of the results (e.g., sex, developmental stage). And I understand not wanting to speculate too much. However, for such life-saving management tools to have a greater chance at being useful in a variety of situations, there must be some deeper

attempt to tie in the methodological choices and outcomes to the animal and its local ecology. For the discussion, you mention reserve size/characteristic as a constraint, but are there recommendations for the use of deterrents given rhino-specific biological, ecological, developmental constraints?

Title: I believe the title still hasn't quite hit the mark. It buries the lede (potential to reduce poaching/disturbance) a bit. I recommend a slight emendation to capture the potential.

Line 58: Change "less" to "fewer"

Lines 61-62: While the statement "increasingly affluent Southeast Asian market" is not untrue, it is shorthand for factors that ultimately drive poaching efforts. And as such, I question the effectiveness of the statement. Although the paper is not about these socio-, cultural-, or politico-economic factors, can this statement be made more true by getting (briefly) at the ultimate drivers?

Line 62: I'm not sure the case for "Consequently" has been made given the previous (lines 60-61) statement. Can you briefly elaborate on the efforts that would lead to rising costs, emotional stress...? I think this brief elaboration would benefit the subsequent information (lines 66 onward). Although this information would be useful, if word count is a concern "Consequently" can be exchanged for a more appropriate adverb.

Comment: Can you speak to the equipment/technology/resources that poachers use to help the reader situate the potential (long-term) effectiveness of eliciting avoidance behaviour via drones, sirens...? Such information would also help the reader to understand the long-term potential for areas of refuge (line 163) to remain as such.

Comment: The inclusion of stats on loss of rhinos via poaching in state/national parks and/vs private reserves would be useful.

Lines 103-107: I think I understand the point here, but the phrasing is a bit passive. Clarity of idea would benefit from more direct syntax. Would encouraging avoidance behaviour not be useful in larger tracts of land? This is elaborated on in the discussion, but if mentioned in the introduction, it needs more explanation (or at the very least a reference to the discussion).

Line 113: Flight TOWARDS a moving person? Hmm? Is this an example of a deterrent?

Lines 120-123: Awkwardly phrased and some grammatical issues; pronoun use and possessive determiners obscure clarity. I recommend rewriting.

Line 172: Perhaps "more slowly"

Line 173: Do you have this information re: stings?

Line 193: Hmm? You mention noise. Is noise awareness the key factor of concern re: drone altitude? Then would you categorize drones as an acoustic deterrent?

Line 200 Change to drone-based.

Lines 208-211: Phrasing is a bit confusing/misleading given that experiment was conducted over a year. As written it implies you had to regularly reclassify individuals. Doesn't entirely clear things up, but a simple fix would be to change "was" to "is" on line 209.

Line 289: Change to "twelve"

Line 301: Remove "during"

Lines 358-361: Clarity would benefit from using active voice.

Line 404: Please read for accuracy and comma placement. How long were ropes soaked in the treatment? When were scents reapplied?

Line 661: Use of "beyond" is confusing as it can refer to distance or time. Do you mean the rhinos continued to move away from the deterrent after the observation period ended?

Line 889: Here you introduce the use of low altitude drones for reconnaissance. Is this in conflict with drones as a means of deterrence? Should there be some discussion as to the potential saturation of drones in an area and the effect of avoidance behaviour?

Decision letter (RSPB-2019-1135.R0)

11-Jun-2019

Dear Mr Penny:

Your revised version of the manuscript "Drones and sirens as a reserve management tool to elicit avoidance behaviour in southern white rhinoceros" has now been peer reviewed and also assessed by an Associate Editor. The (very thorough) comments from the reviewer and the Associate Editor are included at the end of this email for your reference. As you will see, the reviewer and the AE are generally happy with the revisions on the manuscript, but do still have some concerns; we would therefore like to invite you to revise your manuscript to address them.

When revising your manuscript you should also ensure that it adheres to our editorial policies

(<https://royalsociety.org/journals/ethics-policies/>). You should pay particular attention to the following:

Research ethics:

Use of animals and field studies:

Please submit a copy of your revised paper within three weeks. If we do not hear from you within this time your manuscript will be rejected. If you are unable to meet this deadline please let us know as soon as possible, as we may be able to grant a short extension.

Best wishes,

Professor Loeske Kruuk
Editor
mailto: proceedingsb@royalsociety.org

Associate Editor Board Member

Comments to Author:

The authors satisfactorily and thoroughly addressed the editorial and reviewer concerns. In particular, the new introduction provides a stronger contextualization of the scope and impact of the work. The clarification regarding sample size and how it relates to field-specific standards was also beneficial. In addition to several clarification suggestions from the reviewer, my only remaining comments relate to the figures, which I believe require some further attention. 1) To my eyes, the median lines and data points are slightly too small and not standardized (eg, 1a and 1b dots are different sizes), making the interpretation of the plots difficult. Thicker lines and larger dots may improve legibility. 2) I think the relevant figures would benefit from the measure of “awareness” specified in the legends (eg, “duration of investigative, alert and flight behaviours”). 3) The data are jittered (just horizontally?), which should also be specified in the legends. 4) I believe the median line for the siren in Figure 1a is in the wrong place (it does not match up to the displayed data or the statistics reported in the body of the MS itself). Given that this median was likely in the wrong place, please also double check all other plots to ensure they correctly display the statistics.

Reviewer(s)' Comments to Author:

Referee: 1

Comments to the Author(s).

I was pleased to read the revised, much improved MS. I look forward to seeing it published after some further revision. My principal concern still holds: I'm still not seeing a clear, direct connection between rhino biology/physiology, behaviour/ecology and the use of the deterrents (e.g., lines 119-156). It still reads a bit vague. This remains true of the discussion as well, where I see more of a reiteration of results rather than a discussion as to rhino-specific reasons why certain results were observed (e.g., see discussion section on acoustic deterrents). That being said, the authors do include information to explain some of the results (e.g., sex, developmental stage). And I understand not wanting to speculate too much. However, for such life-saving management tools to have a greater chance at being useful in a variety of situations, there must be some deeper attempt to tie in the methodological choices and outcomes to the animal and its local ecology. For the discussion, you mention reserve size/characteristic as a constraint, but are there recommendations for the use of deterrents given rhino-specific biological, ecological, developmental constraints?

Title: I believe the title still hasn't quite hit the mark. It buries the lede (potential to reduce poaching/disturbance) a bit. I recommend a slight emendation to capture the potential.

Line 58: Change “less” to “fewer”

Lines 61-62: While the statement “increasingly affluent Southeast Asian market” is not untrue, it

is shorthand for factors that ultimately drive poaching efforts. And as such, I question the effectiveness of the statement. Although the paper is not about these socio-, cultural-, or politico-economic factors, can this statement be made more true by getting (briefly) at the ultimate drivers?

Line 62: I'm not sure the case for "Consequently" has been made given the previous (lines 60-61) statement. Can you briefly elaborate on the efforts that would lead to rising costs, emotional stress...? I think this brief elaboration would benefit the subsequent information (lines 66 onward). Although this information would be useful, if word count is a concern "Consequently" can be exchanged for a more appropriate adverb.

Comment: Can you speak to the equipment/technology/resources that poachers use to help the reader situate the potential (long-term) effectiveness of eliciting avoidance behaviour via drones, sirens...? Such information would also help the reader to understand the long-term potential for areas of refuge (line 163) to remain as such.

Comment: The inclusion of stats on loss of rhinos via poaching in state/national parks and/vs private reserves would be useful.

Lines 103-107: I think I understand the point here, but the phrasing is a bit passive. Clarity of idea would benefit from more direct syntax. Would encouraging avoidance behaviour not be useful in larger tracts of land? This is elaborated on in the discussion, but if mentioned in the introduction, it needs more explanation (or at the very least a reference to the discussion).

Line 113: Flight TOWARDS a moving person? Hmm? Is this an example of a deterrent?

Lines 120-123: Awkwardly phrased and some grammatical issues; pronoun use and possessive determiners obscure clarity. I recommend rewriting.

Line 172: Perhaps "more slowly"

Line 173: Do you have this information re: stings?

Line 193: Hmm? You mention noise. Is noise awareness the key factor of concern re: drone altitude? Then would you categorize drones as an acoustic deterrent?

Line 200 Change to drone-based.

Lines 208-211: Phrasing is a bit confusing/misleading given that experiment was conducted over a year. As written it implies you had to regularly reclassify individuals. Doesn't entirely clear things up, but a simple fix would be to change "was" to "is" on line 209.

Line 289: Change to "twelve"

Line 301: Remove "during"

Lines 358-361: Clarity would benefit from using active voice.

Line 404: Please read for accuracy and comma placement. How long were ropes soaked in the treatment? When were scents reapplied?

Line 661: Use of "beyond" is confusing as it can refer to distance or time. Do you mean the rhinos continued to move away from the deterrent after the observation period ended?

Line 889: Here you introduce the use of low altitude drones for reconnaissance. Is this in conflict with drones as a means of deterrence? Should there be some discussion as to the potential saturation of drones in an area and the effect of avoidance behaviour?

Author's Response to Decision Letter for (RSPB-2019-1135.R0)

See Appendix B.

Decision letter (RSPB-2019-1135.R1)

27-Jun-2019

Dear Mr Penny

I am pleased to inform you that your manuscript entitled "Using drones and sirens to elicit avoidance behaviour in white rhinoceros as an anti-poaching tactic" has been accepted for publication in Proceedings B. The Associate Editor's comment was 'The authors diligently and thoroughly addressed all remaining comments from myself and the referee', so thank you for your careful revision.

Open Access

Paper charges

Sincerely,

Professor Loeske Kruuk
Editor, Proceedings B
mailto: proceedingsb@royalsociety.org

Associate Editor:
Board Member
Comments to Author:
(There are no comments.)

Appendix A

Author's response

We have listed all comments from the editor, associate editor and the two referees (in red) followed by the measures we took to address them (in black), along with manuscript quotations if appropriate (italics). Line numbers are given for the clean version of the manuscript.

Editor's comments:

As per the Associate Editor's comments as well, please clarify the sample sizes of numbers of animals involved in the different trials. Please also explain how the statistical tests used account for the non-independence of repeated measures on the same animals.

After discussion with the reserve manager, we elected to add sample sizes throughout the methods, given the noted anonymity of the location. A more comprehensive reply is provided in our response to the associate editor's comments below.

The figures also need to give an accurate indication of the data-sets: bar-charts of the form shown are not a helpful way to represent very small samples, but you could add the actual data values on top. (Have a look at Weissgerber et al. 2015. Beyond Bar and Line Graphs: Time for a New Data Presentation Paradigm. PLOS Biology 13:e1002128).

We are grateful to the editor for this suggestion and have replotted the figures to provide greater clarity utilising the format suggested by Weissgerber et al. (2015).

Finally, there needs to be some acknowledgement and discussion of the limitations of the small sample size, which obviously can include a reminder as to why sample sizes are far smaller than typical behavioural studies.

We have commented on sampling limitations found within the study, adding the following text to the discussion:

Lines 311-317:

"The sample sizes (≤ 17 rhinos per analysis) are comparable to existing studies of deterrents [38-40] and reflect the difficulty of exposing free-ranging mammals to experimental stimuli and the need to minimise undue stress [41]. Furthermore, unlike several previous studies [27,32,39], the repeated measures design of the acoustic and drone analyses provide a robust control for individual variation. Perhaps more importantly, the study population and field site are representative of those found in other private reserves [16] where the deterrents will have the greatest conservation impact."

Associate Editor's comments

After receiving two peer reviews, I continue to be impressed by the potential importance of the results presented in the MS: "Drones and sirens reduce poaching risk by eliciting avoidance behaviour in southern white rhinoceros (*Ceratotherium simum simum*)". Understanding mechanisms to reduce poaching risk are of pressing importance and have the potential to deepen knowledge about avoidance and deterrence behavior more generally. However, along with both reviewers, I found the framing the study to be somewhat limited in scope and scalability.

We are grateful to the Associate Editor for their positive response to the submitted manuscript and for seeing the potential importance of this study. On reflection, we agree that our previous version of this manuscript may have been seen as somewhat limited in scope and scalability. As a consequence of the revisions we have made to the manuscript in light of the reviewer's comments below we believe we have addressed this issue.

In addition to the excellent broad points raised by both reviewers regarding contextualizing, framing, and consideration of the poaching impact of the research, I believe that the work would also benefit from the following considerations:

1) The sample size, study subjects, and statistical approach were not properly explained (e.g., how many rhinos were studied should be in the methods; what is the unit of analysis for the models?). Moreover, the statistical methods contain at least one error/typo: The last sentence of the methods reads "all alpha levels were set at 0.05 MULTIPLIED by the number of hypotheses tested." I believe the authors meant to write "...DIVIDED by...".

As mentioned in our previous communications, we had been reluctant to divulge this information in the original manuscript due to security concerns for the study site. However, we recognise the critical importance of providing this information and agree that, by omitting the exact locality of the study site, the security risks are negated. Thus, we have removed the second affiliation of L. MacTavish given that it lists the location of the reserve. We have added the number of rhinos involved in each trial within the appropriate methods sections. For further transparency we have listed the number of observations per analysis in our results section, along with the number of rhinos (N). We have rewritten the data analysis paragraph [from line 220] to clarify the units of analysis and outline how the tests accounted for the repeated measures design. We have also corrected the mistake on alpha levels in the last sentence, rewriting the text to remove ambiguity:

Lines 228-230

"Dunn's tests with Bonferroni corrections were performed on significant results to establish any directions in trend and account for the family-wise error rate."

This clarifies that the alpha level was adjusted only for the post-hoc tests when accounting for multiple comparisons, and that the p-values stated in the tables have already been adjusted to reflect this, without the need for any additional calculations from the reader.

2) Several, more minor issues: (a) I think that the use of multiple acronyms interferes with rather than facilitates comprehension. I would like to see all treatment conditions spelled out in the text. (b) Figure legends should contain more information in general, especially about treatments as well as descriptions of the unit of analysis and the N.

2(a) We utilised acronyms in our previous submission for the sake of brevity. However, we do agree that their utilisation did impact comprehension of the paper. As a result, we have removed all acronyms. Following their introduction as terms 'Cape turtle dove' [CTD] now appears as 'dove' and 'African honey bee' [AHB] as 'bee'. We also now refer to unmanned aerial vehicle [UAV] as 'drone' throughout.

2(b) To address this, we have added sample sizes to all figures and expanded the text to increase precision and clarify the unit of analysis.

3) A detailed response to all the reviewers additional, specific concerns is also needed (e.g., approval by an Ethics Board and if not, please justify).

We are extremely grateful to have received the comments following the careful consideration of our manuscript by the two reviewers. The specific concerns of the reviewers are addressed point by point below. To ensure we still met the length requirements of Proceedings B we also trimmed superfluous information from the manuscript and resubmit the image figure as a supplementary material. However, if space permits and the manuscript is accepted this could be inserted back in if preferred. Beyond wording, no methodological changes were made thus no additional approval was sought from our ethics board. Our previous ethical approval statement still applies and reads as follows: *“Ethical approval was granted by the Animal Welfare and Ethics Review Board of the University of Brighton (Ref: 2018-1127). Project methods were also reviewed and approved by the game reserve managers”*.

If the authors are able to address these concerns, I believe it could be a valuable and unique contribution to the literature.

Response to referee 1

The authors present important data that could be used to help protect rhinos from poaching risk. I think this study is ultimately worthy of publication; the data are sound and presented well, and the topic is pressing. My concern is that the study is framed almost entirely without ecological or behavioural context. I think the paper has the potential to meet the outstanding standards of Proc B if the authors undertake the major revision to restructure, and to a some extent, re-write their paper.

The authors make some strides at providing ecological/behavioural context in the discussion to help explain the results. However, this is belated and insufficient. The authors need to provide more robust justifications for the deterrents they chose, as it relates to the lived experiences of the rhinos. For example, the authors write (starting on line 108), “For the olfactory stimuli, we tested the prediction that rhinos would demonstrate greater avoidance behaviour and reduced investigative behaviour to the scent of chilli oil than to the scent of sunflower oil, due to the potential irritating properties of chilli.” The justification here is devoid of its meaning as it relates to what we know of rhinos.

We understand the reviewer’s concerns and agree that the study should be framed in a stronger ecological and behaviour context. To aid this, we have rewritten the introduction to focus on the behavioural justification behind our choice of each deterrent.

First of all, we state that white rhinos react with flight to disturbances, and that their acute acoustic and olfactory sense may be suitable targets for inciting movement:

Lines 75-81:

*“To date, only one study has reported the use of deterrents in rhino management, in which electric fences were found to be effective at reducing crop raiding in Indian rhinos *Rhinoceros unicornis* in Nepal [19]. White rhinos will also react with flight towards a moving person, but this only consistently occurs at a range of less than 50 m [20]. However, their acute sense of hearing [21] and highly developed sense of smell [22] dispose them towards acoustic and olfactory disturbances [20].”*

We then restructure our examples of other deterrent schemes, specifically relating them to their potential usage with rhinos:

Lines 82-98:

*“Successful deterrents often employ ‘metaphorical’ fences to inhibit animal encroachments into specific areas, through non-physical structures such as sounds and smells [23]. For example, the noise of African honeybees (*Apis mellifera scutellata*) can incite a flight response in African bush elephants *Loxodonta africana* [24]. White rhinos regularly rub themselves against wooden branches and posts [25], behaviour that may provoke defensive swarms of bees, and make them susceptible to its use as an acoustic deterrent. Acoustic deterrents that trigger a generalised threat response (e.g. bangs in rabbits [26]), or act through pain and discomfort (e.g. artificial tones in seals [17]) may also be suitable for use with rhinos. Some deterrents also implement novel technologies into their design (e.g. exposure to drones (unmanned aerial vehicles) in African bush elephants [27]). As the disturbance effects of drones are documented across a wide range of taxa (e.g. bears [28], seals [29], and birds [30]), this has the potential to work in rhinos, aided by their preference for relatively open savannah grasslands [20] a habitat conducive to low-altitude drone flights. Olfactory stimuli such as capsaicin, an irritant present in chillies [31], may also be suitable for deterring rhinos due to its well-documented repelling effect across several species (e.g. in elephants [32-33], monkeys [34] and bears [35]).”*

Furthermore, the authors (starting on line 82) make the very case for the need to provide greater ecological and behavioral information – for the species overall, as well as for the particulars of the study. The authors write, “Poaching risk for rhino populations throughout their distribution is not homogenous [5], being influenced by biophysical [21], geopolitical [22] and socioeconomic factors [23]. Therefore, even at a protected area scale, poaching rate will vary spatially, with poaching hotspots typically centred around geographical features such as waterholes [24] and boundary features [25-26], leading to uneven distribution of anti-poaching efforts [24].” Thus, if any potential deterrents, such as the drones, are to be effectively and situationally deployed to reduce poaching risk, practitioners/managers need to fully understand the results of this study in its broader context. For example, given the complexity of factors under what circumstances would drones be effective. What factors would render such a deterrent useless against poaching threat?

We thank the reviewer for these points and agree they would benefit practitioners. We have rewritten the introduction of our manuscript to frame our research within a broader ecological context, expanding information on the causal inputs of poaching risk. We first remove unneeded information on the socio-economics of poaching and instead place the section outlining poaching risk towards the beginning of the introduction.

We first move the section outlining poaching risk towards the beginning of the introduction. This section has been expanded to more fully outline the problems that can be addressed by deterrents. First we cite from Haas and Ferreira (2017) [9] to emphasise the existence of poaching hotspots and the need for their enhanced protection. Then we cite research by Park et al. (2015) [10] who outlined specific features predictive of rhino poaching (e.g. proximity to roads) then widen this to studies of elephants:

Lines 54-62:

*“Poaching risk for rhino populations throughout their distribution is not homogenous [1], being influenced by biophysical [6], geopolitical [7] and socioeconomic factors [8]. Limited conservation resources are therefore focused on those areas exposed to the greatest levels of poaching risk [9]. Park et al. [10] found an area’s poaching risk to be a function of its distance from the nearest water, buildings, vegetation, and roads, of the number of rhinos present, and of its topography. Spatiotemporal analyses of poaching patterns in African bush elephants (*Loxodonta africana*) show similar results, with the density of conspecifics, roads and rivers, condition of the vegetation, and distance from anti-poaching bases and boundaries all indicators of poaching risk [11-13].”*

Next we cite work by Koen et al. (2017) [14] which identifies how rhino poaching risk can fluctuate over time:

62-67:

Rhino poaching risk is also dependent on the time of day, and phase and position of the moon [14]. Twilight and night are the preferred time of poaching particularly when they coincide with increased levels of lunar illumination [14] which will aid hunting but also poacher interdiction by rangers [9]. Poachers will also take advantage of bad weather conditions, which may limit the scope of patrols and increase their ease of escape [14].

We then reference deterrents as a suitable anti-poaching strategy for controlling this poaching risk in smaller private reserves, including background on the populations found in them:

Lines 68-71:

“The movement of rhinos away from these poaching hotspots may represent a suitable anti-poaching

strategy in private reserves, which are usually fenced and smaller than state or national parks [15-16] but hold approximately 30% of Africa's white rhino population [1]."

We also expand on the conservation implications in our discussion specifically stating the situations most suited to using drones as deterrents:

Lines 415-420:

"Drone deterrents would be most applicable to small private reserves, where rhinos have access to perimeter zones or exposed areas, particularly during heightened periods of risk (e.g. around the full moon [14] or when poaching syndicates are known to be operating in the area [51]). They are less suited for use in larger state or national parks with semi-porous borders and near-constant poaching activity [4]."

Overall, the paper devotes itself to presenting the data, while taking for granted the information (behavioural/ecological/geographical, etc.) that would actually give more heuristic credence to the findings. I look forward to seeing this valuable study presented in the framework it deserves.

Response to referee 2

Comments to the Author(s)

This study brings new information on the potential use of acoustic and olfactory deterrents in anti-poaching efforts to move rhinos away from high-risk areas. The manuscript is well-written. The authors came up with a good idea and went in the footsteps of previous research done mainly on elephants, which showed that various olfactory and acoustic stimuli or unmanned aerial vehicle can deter them from entering unwanted areas (e.g. when wanting to raid crops). The authors tested if these deterrents could also be used to deter rhinos from areas which are considered high-risk because of poaching.

The study was well-designed although it would also be nice to see if the authors had any information on how long approximately the effect of the siren and unmanned aerial vehicle on the rhinos lasted. The authors mentioned that the trials were not repeated on the same individuals sooner than in 24 hours. Does this mean that the individuals can return to the same area at least in a day?

We appreciate the reviewer's comments on experimental design. Unfortunately, as we did not monitor rhino return time we have not commented on this in the manuscript. In our personal observations, the rhinos frequently changed their direction of movement following a disturbance and continued away from the zone they had been in. However, based on more long-term monitoring there were no areas the rhinos avoided. Although future studies would be needed to establish this.

The 24 hour period was selected as a precaution to overly avoid the animals during the experimentation stage. While the rhinos could return in this time frame, we do not recommend any limit to their frequency of exposure as a method to reduce poaching risk.

In addition, the distance to which the rhinos ran away from the siren was not large (median 46m for all rhinos altogether, but only 3m for adult bulls). This distance is short taking into account that the poachers can hit anytime and it is not such a problem for them to move and search for the rhinos. However, the drones have a potential to pursue the rhinos for longer distances and could be used to get them out of high-risk areas in smaller reserves during the most sensitive time like e.g. around the full moon which the poachers are known to prefer (e.g. Mulero-Pazmany et al. 2014).

We thank the reviewer for the comments and agree this needs expansion. We have now added information to the discussion to emphasise that although the distances moved from the deterrent appear short, the responses were only monitored for one minute and thus show potential to be greatly expanded:

Lines 304-307:

"Although the distances travelled were short, in many cases the rhinos continued to flee beyond the duration of the monitored periods, with flight distances of over 500 metres observed on two occasions in response to the drone, even without pursuit, and distances of over 250 m travelled from the siren."

In our introduction we now greatly expand on the reasons for variations in poaching risk and how they result in poaching hotspots. Lines 54-67. We also add a section to the discussion to highlight the situations where the drone is appropriate for use:

Lines 415-418

"Drone deterrents would be most applicable to small private reserves, where rhinos have access to perimeter zones or exposed areas, particularly during heightened periods of risk (e.g. around the full moon [14] or when poaching syndicates are known to be operating in the area [51]). They are less

suited for use in larger state or national parks with semi-porous borders and near-constant poaching activity [4].”

Here are some specific comments and a few suggestions that may improve the manuscript:

I think that the name of the article should be changed as the authors did not test if the drones and sirens reduce the risk of poaching. They tested if the rhinos move away from the drones and the sound of siren, but it remains unknown whether this can reduce the risk of poaching especially since it seems that the rhinos can come back to the same area soon and the distance to which they move away from the siren is short.

We agree with the reviewer and have changed the name to reflect this. The paper is now entitled:

- Drones and sirens as a reserve management tool to elicit avoidance behaviour in southern white rhinoceros (*Ceratotherium simum simum*).

In this line, I also suggest to re-phrase the last sentence of the abstract (line 40-42).

We have rephrased this to avoid mention of poaching risk and it now reads:

Lines 35-37:

“Our findings indicate that deterrents are a prospective low-cost and in situ method to manage rhino movement in game reserves.”

Line 146–152: You write that the playbacks were played when the rhinos were in distances 50–150 m from the speaker. It seems that the loudness of the playbacks was not changed depending on various distances of rhinos from the speakers ? Do you think that this could have had any effect on the reactions of rhinos? That those who were closer reacted more intensively? Do you think that this could have had any effect on the reactions of rhinos? That those who were closer reacted more intensively?

The reviewer is correct in their assertion. While we standardised the volume as much as possible by only broadcasting the sounds within a 100 m range we did not change it depending on each rhino’s distance. There would also have been minor variations in amplitude brought on by slight differences in vegetation, terrain and wind speed despite our efforts to standardise them. Despite this, we do not believe adjusting the amplitude will have had a major impact on the results, as the playbacks were audible to the rhinos at both the minimum and maximum distances tested. The differences in volume may have presented a problem if we had recorded observations for longer than 1-minute, as the stimulus will have attenuated to zero as the rhinos moved further away. As this was not done, the strength of a continued behavioural response will not have affected our results.

Line 150: I am missing the information about the manufacturer and parameters of the horn speakers. Their frequency range, etc.

The missing information on manufacturer and their parameters has been added:

Lines 156-157:

“Sounds were broadcast through two 30 watt horn speakers (frequency range: 250 Hz - 10 kHz; TOA)”

Line 153: Could you, please, include here why you did not continue recording the rhino behaviour after the end of the playback? It would be interesting to know the reactions of rhinos even after the

playback ended and to see what effect it had on them. Did the rhinos always stop reacting with either awareness /moving away already during the playback? Or was it impossible for you to continue with the observations due to e.g. the subjects getting out of sight?

The reviewer was correct in their assumptions, as noted above we now comment on their continued movement in our discussion on lines 304-307. We now also state that the minute period was chosen to increase comparability with the drone deterrent. To address this and make it clear to the reader we added the following sentence the methods:

Lines 161-163:

“Observations were truncated at one minute to ensure that rhinos remained visible throughout the experiment and to aid their comparability with data taken from the drone.”

Line 158-159: range finder info? Manufacturer, etc. ?

The missing information on model and manufacturer has been added:

Lines 167-168:

“via a range finder (Leica Rangemaster CRF 1600-R).”

Line 243-245: Since you also analysed the data based on sorting out the individuals into sex-age classes, there should be an explanation and a citation (e.g. Owen-Smith (1973)) in the methods how you identified individuals as adults/subadults. In bulls, this could differ depending on if you regard them as adults based either only on their sexual maturity or on their socio-sexual maturity when they become solitary between the age 10–12 years.

We thank the reviewer for this point and agree it would increase behavioural understanding for the reader. To address this we have added the following information to our methods:

Lines 133-136:

“Rhinos were classed as subadults from maternal independence until they reached socio-sexual maturity. This was when males become solitary and/or territorial at 10-12 years old and at around 7 years old in females after the birth of their first calf) [20].”

Figures: Please, describe in captions what the boxes and the measures of central tendency represent.

We apologise for this omission. Instead of expanding the captions as described we have now replotted the figures without the boxes, to make the entire spread of the dataset visible. We followed guidelines by Weissgerber et al. 2015 as suggested by the editor.

Figures 1, 3, 4, 5: Some of the medians cannot be seen due to the black boxes. Can you, please, make the boxes lighter so the medians could also be seen in the figures?

This comment is also met by replotting the graphs to show the entire dataset.

Appendix B

Author's response

We have listed all comments from the associate editor and the referee (in red) followed by the measures we took to address them (in black), along with manuscript quotations if appropriate (italics). The line numbers we provide are for the clean version of the manuscript.

We have also appended a table outlining how the line numbers provided by the reviewer correspond to the clean version of the previously submitted manuscript.

Associate Editor's comments:

The authors satisfactorily and thoroughly addressed the editorial and reviewer concerns. In particular, the new introduction provides a stronger contextualization of the scope and impact of the work. The clarification regarding sample size and how it relates to field-specific standards was also beneficial. In addition to several clarification suggestions from the reviewer, my only remaining comments relate to the figures, which I believe require some further attention. 1) To my eyes, the median lines and data points are slightly too small and not standardized (eg, 1a and 1b dots are different sizes), making the interpretation of the plots difficult. Thicker lines and larger dots may improve legibility. 2) I think the relevant figures would benefit from the measure of "awareness" specified in the legends (eg, "duration of investigative, alert and flight behaviours"). 3) The data are jittered (just horizontally?), which should also be specified in the legends. 4) I believe the median line for the siren in Figure 1a is in the wrong place (it does not match up to the displayed data or the statistics reported in the body of the MS itself). Given that this median was likely in the wrong place, please also double check all other plots to ensure they correctly display the statistics.

Response to Associate Editor:

We are grateful to the associate-editor for noticing the above issues with the plots.

1. We have increased point size and line thickness throughout all plots to improve legibility and ensured they are standardised.
2. We have added a description of awareness to the figure legends.
3. We have specified in the legends that the points are horizontally jittered.
4. The median lines for Figure 1A were misaligned and have been corrected. The mistake was in the figure script and did not affect analysis. We checked the medians throughout but they required no other changes.

Reviewer's Comments:

I was pleased to read the revised, much improved MS. I look forward to seeing it published after some further revision. My principal concern still holds: I'm still not seeing a clear, direct connection between rhino biology/physiology, behaviour/ecology and the use of the deterrents (e.g., lines 119-156). It still reads a bit vague. This remains true of the discussion as well, where I see more of a reiteration of results rather than a discussion as to rhino-specific reasons why certain results were observed (e.g., see discussion section on acoustic deterrents). That being said, the authors do include information to explain some of the results (e.g., sex, developmental stage). And I understand not wanting to speculate too much. However, for such life-saving management tools to have a greater chance at being useful in a variety of situations, there must be some deeper attempt to tie in the methodological choices and outcomes to the animal and its local ecology. For the discussion, you

mention reserve size/characteristic as a constraint, but are there recommendations for the use of deterrents given rhino-specific biological, ecological, developmental constraints?

We thank the reviewer for their positive comments. To address these concerns we have rewritten parts of manuscript to place the focus specifically on how the deterrents relate to rhino behaviour, physiology and ecology.

For the deterrents-section of the introduction we have changed the focus from successful usage in other species towards rhino behaviour. We hope that the reasoning behind our methodological choices is now much clearer. These changes can be found on lines 82 to 104 and read as follows:

*“In other species, successful deterrents employ non-physical structures such as sounds and smells to create ‘metaphorical fences’ [20]. White rhinos’ disposition towards acoustic and olfactory disturbances [21] may mean they are also susceptible to these forms of deterrent. Acoustic deterrents can elicit a generalised threat response through loud or novel noises (e.g. bangs in rabbits [22]) or repel animals through pain and discomfort (e.g. artificial tones in seals [17]). The broadcast of such stimuli could exploit white rhinos’ acute sense of hearing [23] to inhibit animal encroachments into specific areas. Other acoustic deterrents rely on conditioned responses towards unpleasant experiences [24]. White rhinos regularly disturb vegetation when rubbing against branches and moving through scrub [21], behaviour which may provoke defensive swarms of African honeybees (*Apis mellifera scutellata*) [25]. The broadcast of bee noise has the potential to incite a flight response in rhinos, if as occurs in African bush elephants *Loxodonta africana* [24], individuals have experienced past aversive conditioning to stings. Olfactory stimuli such as capsaicin, an irritant present in chillies [26], have well-documented repelling effect across several species (e.g. in elephants [27-28], monkeys [29] and bears [30]). White rhinos’ have a highly developed sense of smell [31] and so exposure to noxious or novel scents could trigger avoidance behaviours through pain or neophobia. White rhinos’ will also flee from helicopters [32] and their preference for relatively open savannah grasslands [33] make habitats conducive to aerial pursuit. Hahn et al. [34] demonstrated how the disturbance effects of drones can be used to repel African bush elephants. Given that drones incite similar avoidance behaviours in a wide range of taxa (e.g. bears [35], seals [36], and birds [37]), the potential exists for them to incite a flight response in white rhinos.”*

References added:

32. Lindsey, P.A. & Taylor, A., 2011 A study on the dehorning of African rhinoceroses as a tool to reduce the risk of poaching. South African Department of Environmental Affairs, Johannesburg.
33. Shrader, A., Owen-Smith, N. & Ogotu, J. 2006 How a mega-grazer copes with the dry season: food and nutrient intake rates by white rhinoceros in the wild. *Functional Ecology* 20, 376-384. (doi:10.1111/j.1365-2435.2006.01107.x)

To address the concerns around the discussion, we have rewritten all sections to reduce the repetition of results and align the focus to rhino biology and ecology. We have also edited the work to be concise and keep within the word limit.

For example, we now suggest that the lack of a flight response to bees was due to the thickness of rhino skin and include a reference on the toughness of its structure:

Lines 330-334: *“The rhinos did not appear to perceive the risk from the bees as great enough to initiate flight behaviour [40]. This contrasts with the growing body of evidence for their use with African bush*

elephants [24-25,45], perhaps because the thick skin of rhinos, adapted to shield against attacks from conspecifics, provides sufficient protection against aggressive swarms of bees [46].”

Reference added:

46. Shadwick, R.E., Russell, A.P. & Lauff, R.F. 1992 The structure and mechanical design of rhinoceros dermal armour. *Philosophical Transactions of the Royal Society of London. Series B: Biological Sciences* 337, 419-428. (doi:10.1098/rstb.1992.0118)

We also now address the constraints of deterrents in terms of differences in reserve ecology and rhino socio-behaviour:

Lines 314-340: *“Rhinos may soon return to an area if the costs of avoiding the stimulus are outweighed by the benefits of staying put and foraging [40]. Thus, the chances of return could be lowered if higher quality areas of habitat are maintained in other more suitable areas of a reserve. Deterrents may be less effective in periods of reduced resource availability, such as drought, when rhinos have a more limited choice in grazing areas [33]. Dominant males will also be more inclined to return to an area than other social classes, given their need to regularly patrol and demarcate their territories [21,31].”*

33. Shrader, A., Owen-Smith, N. & Ogotu, J. 2006 How a mega-grazer copes with the dry season: food and nutrient intake rates by white rhinoceros in the wild. *Functional Ecology* 20, 376-384. (doi:10.1111/j.1365-2435.2006.01107.x)

Title: I believe the title still hasn't quite hit the mark. It buries the lede (potential to reduce poaching/disturbance) a bit. I recommend a slight emendation to capture the potential.

We have changed the title to: 'Using drones and sirens to elicit avoidance behaviour in white rhinoceros as an anti-poaching tactic'.

Line 58: Change “less” to “fewer”

We have changed this in the manuscript.

Lines 61-62: While the statement “increasingly affluent Southeast Asian market” is not untrue, it is shorthand for factors that ultimately drive poaching efforts. And as such, I question the effectiveness of the statement. Although the paper is not about these socio-, cultural-, or politico-economic factors, can this statement be made more true by getting (briefly) at the ultimate drivers?

To address these concerns, we have added detail on why horn is desirable and added a reference to where the reader can find further information on the subject as we agree with the reviewer that this is largely outside the scope of this manuscript.

Lines 48-50: *“However, this success is threatened by a rapid increase in rhino poaching [1] fuelled by a surge in demand from an increasingly affluent Southeast Asian market [4], where horn is used medicinally and as a symbol of status [5].”*

Reference added:

5. Truong, V., Dang, N. & Hall, C. 2015 The marketplace management of illegal elixirs: illicit consumption of rhino horn. *Consumption Markets & Culture* 19, 353-369. (doi:10.1080/10253866.2015.1108915)

Line 62: I'm not sure the case for "Consequently" has been made given the previous (lines 60-61 [48-49?]) statement. Can you briefly elaborate on the efforts that would lead to rising costs, emotional stress...? I think this brief elaboration would benefit the subsequent information (lines 66 [54] onward). Although this information would be useful, if word count is a concern "Consequently" can be exchanged for a more appropriate adverb.

We have addressed these concerns by describing typical anti-poaching methods along with our comment on costs. Given that we are near the maximum word count we have elected to remove the statement on emotional stress, as the financial costs are our main point of emphasis:

Lines 50-55: *"The rising costs of effective anti-poaching security are putting significant financial pressure on both national parks and private reserves [3], where the apprehension of poachers and reducing incursions are primarily achieved through foot and vehicle patrols [6]. There is thus a clear need to identify effective, low-cost and readily applicable techniques to aid on-the-ground conservation efforts."*

Comment: Can you speak to the equipment/technology/resources that poachers use to help the reader situate the potential (long-term) effectiveness of eliciting avoidance behaviour via drones, sirens...? Such information would also help the reader to understand the long-term potential for areas of refuge (line 163) to remain as such.

There is wide a variation in poaching tactics depending on whether it's carried out by an individual or orchestrated by an organised criminal syndicate. Although there are reports of increasingly sophisticated rhino poaching gangs (e.g. instances with helicopters), the technologies poachers' use are generally restricted to firearms, GPS and mobile phones (e.g. see comment by Duffy 2014) with poachers typically operating on foot (Haas and Ferreira, 2018). Reserves' are likely to continue to experience differential levels of risk in space and time (leading to zones of relative refuge and danger) regardless of any technological changes. While we agree this knowledge would be useful to the reader we find it difficult to speculate on future changes in poacher capabilities and do not believe we can provide a useful assessment of future trends. Thus, we have not included this information.

Duffy, R., 2014. Waging a war to save biodiversity: the rise of militarized conservation. *International Affairs*, 90(4), pp.819-834.

Haas, T.C. and Ferreira, S.M., 2018. Optimal patrol routes: Interdicting and pursuing rhino poachers. *Police Practice and Research*, 19(1), pp.61-82.

Comment: The inclusion of stats on loss of rhinos via poaching in state/national parks and/vs private reserves would be useful.

The South African Department of Environmental Affairs has resumed releasing information on poaching rates across different provinces (see link), but we are unaware of any recent stats available on the private reserve poaching rate (at least publically listed) that we could cite. This includes from non-governmental compilers of poaching stats (e.g. stoprhinopoaching.com/)

https://www.environment.gov.za/mediarelease/mokonyane_2018integratedstrategic_management_ofrhinoceros_2019feb

As a result we have not been able to include these in the revised manuscript, as per the reviewers request.

Lines 103-107: I think I understand the point here, but the phrasing is a bit passive. Clarity of idea would benefit from more direct syntax. Would encouraging avoidance behaviour not be useful in larger tracts of land? This is elaborated on in the discussion, but if mentioned in the introduction, it needs more explanation (or at the very least a reference to the discussion).

As suggested, we have made the phrasing more direct and now avoid implying that the strategy is only suited for use in private reserves. Instead we now state that it is 'most suited' for use in these locations. Additionally, the rewrite now makes it clear that the presence of a fenced perimeter, coupled with more hands-on management are partly what makes such an approach more appropriate to these localities.

Lines 70-74: *"The movement of rhinos away from these poaching hotspots could be a useful anti-poaching tactic. Such a strategy would be most suited for use in private reserves, which are usually fenced and smaller than state or national parks [15-16] but hold approximately 30% of Africa's white rhino population and these animals are typically subject to more intensive management than national park populations [1]."*

Line 113: Flight TOWARDS a moving person? Hmm? Is this an example of a deterrent?

This statement was no longer relevant to our rewritten introduction and has been removed. Our original intention was to state that the flight behaviour was exhibited towards the person, rather than the rhino moving in the direction of the person. A less ambiguous statement would have read: *"White rhinos will also react with flight in response to a moving person, but this only consistently occurs at a range of less than 50 m."* The reviewer is correct that this could be considered a form of deterrent but we haven't seen it reported as such in a study.

Lines 120-123: Awkwardly phrased and some grammatical issues; pronoun use and possessive determiners obscure clarity. I recommend rewriting.

This section referred to the information on bees given in the introduction and has been rewritten to increase clarity and emphasise the biology of the rhinos. See lines 90-95.

Line 172: Perhaps "more slowly"

This sentence was removed in the rewrite.

Line 173: Do you have this information re: stings?

Given the observed presence of bees throughout the reserve and rhino longevity (one had resided on the reserve for over twenty years) coupled with their tendency to crash through vegetation, there is a high likelihood they were exposed to bees, although we did not observe this. There are also several studies that have observed stinging is frequent enough in elephants, which share similar habits and habitats to rhinos, to result in a conditioned response (King 2010).

25. King, L., 2010 The interaction between the African elephant (*Loxodonta africana africana*) and the African honey bee (*Apis mellifera scutellata*) and its potential application as an elephant deterrent. Doctoral dissertation, University of Oxford.

Line 193: Hmm? You mention noise. Is noise awareness the key factor of concern re: drone altitude? Then would you categorize drones as an acoustic deterrent?

Although we primarily link the response to sound, rhinos were observed scanning their heads in response to the disturbance as well as their ears, indicating there may be a visual as well as acoustic

element to their reaction. This is particularly true in other species e.g. the shape and wing profile of drones influences the response of waterfowl (possibly because it mimics raptors – see McEvoy, 2016). We could label it an audio-visual stimuli but feel drone is a better descriptor.

McEvoy, J.F., Hall, G.P. and McDonald, P.G., 2016. Evaluation of unmanned aerial vehicle shape, flight path and camera type for waterfowl surveys: disturbance effects and species recognition. PeerJ, 4, p.e1831.

Line 200 [Line 125]: Change to drone-based.

We have changed this in the manuscript.

Lines 208-211: Phrasing is a bit confusing/misleading given that experiment was conducted over a year. As written it implies you had to regularly reclassify individuals. Doesn't entirely clear things up, but a simple fix would be to change "was" to "is" on line 209.

We agree this was confusing. In the interests of word count we have followed the reviewer's suggestion and changed 'was' to 'is'.

Line 289: Change to "twelve"

We have changed this in the manuscript.

Line 301: Remove "during"

We have changed this in the manuscript.

Lines 358-361: Clarity would benefit from using active voice.

We believe this comment referred to lines 239-242 in the previously submitted version of the clean manuscript and have rectified the prose accordingly. It now reads:

Lines 243-245: *"Significantly longer durations of awareness were exhibited in response to the siren (median = 57.5 sec) than to either the bee (median = 8.5 sec) or dove (median = 0 sec) treatments (Friedman χ^2 (2) = 15.591, $p < 0.001$, $n = 12$, obs. = 36; Figure 1; Table 2)."*

Line 404: Please read for accuracy and comma placement. How long were ropes soaked in the treatment? When were scents reapplied?

We have added the missing information on how long the ropes were infused with scent for:

"Lengths of 5 m natural fibre sisal rope were infused with scent by soaking them in one of the two treatment types for 24 hours. Deployed ropes had scents reapplied after five days."

Line 661: Use of "beyond" is confusing as it can refer to distance or time. Do you mean the rhinos continued to move away from the deterrent after the observation period ended?

We have rephrased this to indicate the rhinos moved both after the monitored period had ended and reached greater distances than those reported in the results:

"Although the distances travelled were short, in many cases rhinos continued to flee after observations had ended; on two occasions rhinos ran over 500 metres in response to the drone, even without pursuit, and on four occasions, over 250 metres from the siren."

Line 889: Here you introduce the use of low altitude drones for reconnaissance. Is this in conflict with drones as a means of deterrence? Should there be some discussion as to the potential saturation of drones in an area and the effect of avoidance behaviour?

The reconnaissance flights we refer to are flown at high enough altitudes to avoid disturbing rhinos (100-180 m), as shown by our own results and the study cited. As the noise and visual impact at these higher altitudes are negligible, the use of drones for reconnaissance are unlikely to have a significant impact on habituation. However, as we outline earlier on in the discussion [lines 344 to 366], our results do not rule out that habituation will occur after continued use. Thus, more research is needed to see how long and over what time scales flight responses can be maintained.

Table 1: The line numbers provided to us did not appear to line up with our versions of either the clean or tracked versions of the manuscript. Thus, we have provided a reference table listing how the comments (in red) correspond to the clean version of the previously submitted manuscript (in black).

Referee 1		
I was pleased to read the revised, much improved MS. I look forward to seeing it published after some further revision. My principal concern still holds: I'm still not seeing a clear, direct connection between rhino biology/physiology, behaviour/ecology and the use of the deterrents (e.g., lines 119-156 = 84-121). It still reads a bit vague. This remains true of the discussion as well, where I see more of a reiteration of results rather than a discussion as to rhino-specific reasons why certain results were observed (e.g., see discussion section on acoustic deterrents). That being said, the authors do include information to explain some of the results (e.g., sex, developmental stage). And I understand not wanting to speculate too much. However, for such life-saving management tools to have a greater chance at being useful in a variety of situations, there must be some deeper attempt to tie in the methodological choices and outcomes to the animal and its local ecology. For the discussion, you mention reserve size/characteristic as a constraint, but are there recommendations for the use of deterrents given rhino-specific biological, ecological, developmental constraints?		
Title: I believe the title still hasn't quite hit the mark. It buries the lede (potential to reduce poaching/disturbance) a bit. I recommend a slight emendation to capture the potential.		
Reviewers Line Numbers for specific comments	Revision	What lines we believe the reviewer to be referring to in clean version of the previous manuscript.
Line 58	Change "less" to "fewer"	Line 46
Lines 61-62	While the statement "increasingly affluent Southeast Asian market" is not untrue, it is shorthand for factors that ultimately drive poaching efforts. And as such, I question the effectiveness of the statement. Although the paper is not about these socio-, cultural-, or politico-economic factors, can this statement be made more true by getting (briefly) at the ultimate drivers?	Lines 49-50
Line 62	I'm not sure the case for "Consequently" has been made given the previous (lines 60-61 = 48-49) statement. Can you briefly elaborate on the efforts that would lead to rising costs, emotional stress...? I think this brief elaboration would benefit the subsequent information (lines 66 =	Line 50

	54 onward). Although this information would be useful, if word count is a concern “Consequently” can be exchanged for a more appropriate adverb.	
Comment: Can you speak to the equipment/technology/resources that poachers use to help the reader situate the potential (long-term) effectiveness of eliciting avoidance behaviour via drones, sirens...? Such information would also help the reader to understand the long-term potential for areas of refuge (line 163 = 106) to remain as such.		
Comment: The inclusion of stats on loss of rhinos via poaching in state/national parks and/vs private reserves would be useful.		
Lines 103-107	I think I understand the point here, but the phrasing is a bit passive. Clarity of idea would benefit from more direct syntax. Would encouraging avoidance behaviour not be useful in larger tracts of land? This is elaborated on in the discussion, but if mentioned in the introduction, it needs more explanation (or at the very least a reference to the discussion).	Lines 68-73
Line 113	Flight TOWARDS a moving person? Hmm? Is this an example of a deterrent?	Line 78
Lines 120-123	Awkwardly phrased and some grammatical issues; pronoun use and possessive determiners obscure clarity. I recommend rewriting.	Lines 85-88
Line 172	Perhaps “more slowly”	Line 114
Line 173	Do you have this information re: stings?	Line 115
Line 193	Hmm? You mention noise. Is noise awareness the key factor of concern re: drone altitude? Then would you categorize drones as an acoustic deterrent?	Line 118
Line 200	Change to done-based.	Line 125
Lines 208-211	Phrasing is a bit confusing/misleading given that experiment was conducted over a year. As written it implies you had to regularly reclassify individuals. Doesn't entirely clear things up, but a simple fix would be to change “was” to “is” on line 209.	Line 133-135
Line 289	Change to “twelve”	Line 171
Line 301	Remove “during”	Line 183
Lines 358-361	Clarity would benefit from using active voice.	Lines 239-242 (uncertain of this)

Line 404	Please read for accuracy and comma placement. How long were ropes soaked in the treatment? When were scents reapplied?	Lines 209-211
Line 661	Use of "beyond" is confusing as it can refer to distance or time. Do you mean the rhinos continued to move away from the deterrent after the observation period ended?	Line 305
Line 889	Here you introduce the use of low altitude drones for reconnaissance. Is this in conflict with drones as a means of deterrence? Should there be some discussion as to the potential saturation of drones in an area and the effect of avoidance behaviour?	Line 409